# Neuron-Enhanced AutoEncoder Matrix Completion and Collaborative Filtering: Theory and Practice

**Jicong Fan**[1,2]    **Rui Chen**[1,2]    **Zhao Zhang**[3]*    **Chris H.Q. Ding**[1]
[1]School of Data Science, The Chinese University of Hong Kong, Shenzhen, China
[2]Shenzhen Research Institute of Big Data, Shenzhen, China
[3]School of Computer Science & Information Engineering, Hefei University of Technology, China
`fanjicong@cuhk.edu.cn  116010018@link.cuhk.edu.cn`
`cszzhang@gmail.com  chrisding@cuhk.edu.cn`

## Abstract

Neural networks have shown promising performance in collaborative filtering and matrix completion but the theoretical analysis is limited and there is still room for improvement in terms of the accuracy of recovering missing values. This paper presents a **n**euron-**e**nhanced **a**utoencoder **m**atrix **c**ompletion (AEMC-NE) method and applies it to collaborative filtering. Our AEMC-NE adds an element-wise autoencoder to each output of the main autoencoder to enhance the reconstruction capability. Thus it can adaptively learn an activation function for the output layer to approximate possibly complicated response functions in real data. We provide theoretical analysis for AEMC-NE as well as AEMC to investigate the generalization ability of autoencoder and deep learning in matrix completion, considering both missing completely at random and missing not at random. We show that the element-wise neural network has the potential to reduce the generalization error bound, the data sparsity can be useful, and the prediction performance is closely related to the difference between the numbers of variables and samples. The numerical results on synthetic data and five benchmark datasets demonstrated the effectiveness of AEMC-NE in comparison to many baselines.

## 1 Introduction

Recommendation systems (Adomavicius & Tuzhilin, 2005) aim to provide personalized recommendations based on various information such as user purchase records, social networks, user features, and item (or product) features. One important technique used by recommendation systems is collaborative filtering (Billsus & Pazzani, 1998; Mnih & Salakhutdinov, 2008; Koren et al., 2009; Zhang et al., 2019) based on matrix factorization or matrix completion (Srebro & Shraibman, 2005; Candès & Recht, 2009; Hu et al., 2012; Vandereycken, 2013; Shamir & Shalev-Shwartz, 2014; Chen et al., 2014; Sun & Luo, 2015; Nie et al., 2015; Fan et al., 2019; Fan, 2022b). In collaborative filtering (CF) problems or recommendation problems more generally, the rating matrices are usually highly incomplete (Resnick et al., 1994; Breese et al., 1998; Koren et al., 2009) and the missing rates are as high as 0.95, due to the naturally scarce interaction between users and items. Matrix factorization or completion based CF methods usually exploit the potential low-rank structure of the incomplete rating matrix to recover the missing values. The low-rankness can be obtained by low-rank factorization (Koren et al., 2009), nuclear norm minimization (Candès & Recht, 2009), or Schatten-$p$ quasi norm minimization (Fan et al., 2019). Particularly, Lee et al. (2016) proposed a method LLORMA that approximates the rating matrix as a weighted sum of a few low-rank matrices. LLORMA outperformed low-rank matrix completion methods in collaborative filtering, which indicates that rating matrices in real applications may have more complicated structures rather than a single low-rank structure. There have been a few works of high-rank matrix completion (Eriksson et al., 2012; Yang et al., 2015; Elhamifar, 2016; Fan & Chow, 2017; Ongie et al., 2017; Fan & Udell, 2019; Fan et al., 2020; Fan, 2022a). These works often assume that the columns of a given data matrix are generated

---
*Corresponding author

from a union of subspaces or low-dimensional manifolds. However, these works haven't shown the effectiveness of high-rank matrix completion methods in CF.

The success of neural networks and deep learning in computer vision and natural language processing inspired researchers to design neural networks for CF (Salakhutdinov et al., 2007; Dziugaite & Roy, 2015; Sedhain et al., 2015; Wu et al., 2016; Zheng et al., 2016; He et al., 2017; van den Berg et al., 2017; Fan & Cheng, 2018; Yi et al., 2020). For instance, Salakhutdinov et al. (2007) proposed a restricted Boltzmann machines (Hinton et al., 2006) based CF method called RBM-CF, which showed high performance in the Netflix challenge (Bennett & Lanning, 2007). Sedhain et al. (2015) proposed AutoRec, an autoencoder (Hinton & Salakhutdinov, 2006; Bengio et al., 2007) based CF method, which predicts unknown ratings by an encoder-decoder model $\hat{x} = W_2\sigma(W_1x)$, where $x$ denotes the incomplete ratings on one item or of one user and $W_1, W_2$ are weight matrices to optimize. AutoRec can be regarded as an autoencoder-based matrix completion (AEMC) method. AutoRec, or AEMC equivalently, outperformed LLORMA slightly on several benchmark datasets (Sedhain et al., 2015). Zheng et al. (2016) proposed a method called CF-NADE, in which parameters are shared between different ratings, and it achieved promising performance in several benchmarks. He et al. (2017) proposed to use a neural network to learn the interaction function or similarity between users and items. The method showed some improved performance over the baselines in the setting of implicit feedback (Rendle et al., 2009) but its performance in the setting of explicit feedback of benchmark datasets is still unknown even today (Rendle et al., 2020; Xu et al., 2021). Muller et al. (2018) proposed a kernel-based reparametrized neural network, in which the weight between two units is set to be a weighted kernel function of the location vectors. The method works well in data visualization and collaborative filtering. Interestingly, Yi et al. (2020) found that the expected value of the output layer of a neural network depends on the sparsity of the input data. They proposed a simple yet effective method called sparsity normalization to improve the performance of neural networks with sparse input data such as the highly incomplete rating matrices.

It is worth mentioning that existing autoencoder-based CF or matrix completion methods such as (Sedhain et al., 2015; Wu et al., 2016; Muller et al., 2018; Yi et al., 2020) use linear activation function in the output of the decoder, i.e., $\hat{x} = W_L h_{L-1}$, where $W_L$ denotes the weights of the output layer and $h_{L-1}$ denotes the features given by the last hidden layer. Thus, these methods are under the assumption that the ratings are linear interactions between user features and item features, though the features can be nonlinear. Such an assumption may not be true or not optimal in real problems, especially when the data are bounded (e.g. images) or are collected by sensors (e.g. medical and chemical sensors) with nonlinear response functions. We suspect that the rating values given by users on items are from some nonlinear response functions because humans have complex emotion or decision curves (LeDoux, 2000; Baker, 2001). A naive method to incorporate nonlinear interaction is using nonlinear activation functions such as the sigmoid function (with rescaling) in the output layer of the decoder, which however has much lower performance than using a linear activation function. That's why existing autoencoder-based CF methods use only linear activation function. Note that a pre-specified activation function for the output layer of the decoder may work on specific data but may be far away from the possible optimal choice.

Note that collaborative filtering is a special case of the general missing data imputation problem that also covers many other tasks such as image or video inpainting and missing value imputation as a preprocessing step for further data analysis. Although neural networks and deep learning have shown promising performance in collaborative filtering and other missing data imputation tasks (Yoon et al., 2018; Mattei & Frellsen, 2019), the theoretical analysis is very limited. It is very necessary and crucial to provide theoretical guarantees for neural networks and deep learning based collaborative filtering and missing data imputation.

**Contribution.** We present a novel matrix completion method AEMC-NE, which is an enhanced autoencoder approach to matrix completion and collaborative filtering. AEMC-NE is composed of two different neural networks, one is an autoencoder to reconstruct the incomplete rating matrix, while the other is an element-wise neural network to learn an activation function adaptively for the output layer of the first autoencoder. We provide theoretical analysis for AEMC-NE as well as AEMC, considering both missing completely at random and missing not at random. Specifically, we prove that the element-wise neural network has the potential to reduce the upper bound of the prediction error for the unknown ratings. We also prove that the data sparsity is not problematic but useful. Further, we demonstrate the effectiveness of our AEMC-NE on synthetic data and five benchmarks including *MovieLens-100k*, *MovieLens-1M*, *MovieLens-10M*, *Douban*, and *Flixster*.

## 2 NEURON-ENHANCED AEMC

Suppose we have an incomplete data matrix $\boldsymbol{X} = (\boldsymbol{x}_1, \boldsymbol{x}_2, \ldots, \boldsymbol{x}_n) \in \mathbb{R}^{m \times n}$ (e.g. a rating matrix), where $m$ is the number of variables, $n$ is the number of samples, and the index set of observed entries is denoted as $S$. Our goal is to predict the missing entries of $\boldsymbol{X}$, i.e., $\{X_{ij} : (i,j) \in [m] \times [n] \backslash S\}$. One approach is to fill the missing entries with some values (e.g., mean or zero) temporarily and then construct an imputation model. Specifically, let $\tilde{\boldsymbol{X}}$ be the temporarily imputed matrix, where $\tilde{X}_{ij} = X_{ij}$ for all $(i,j) \in S$, we learn a nonlinear function $f : \mathbb{R}^m \to \mathbb{R}^m$ such that $\sum_{i=1}^{n} \left\| \boldsymbol{s}_i \odot (\tilde{\boldsymbol{x}}_i - f(\tilde{\boldsymbol{x}}_i)) \right\|^2$ is as small as possible, where $\boldsymbol{s}_i$ is a binary vector denoting whether the corresponding element in $\boldsymbol{x}_i$ is observed or not. The problem is formulated as

$$\underset{f \in \mathcal{F}}{\text{minimize}} \left\| \boldsymbol{S} \odot (\tilde{\boldsymbol{X}} - f(\tilde{\boldsymbol{X}})) \right\|_F^2 \tag{1}$$

where $\boldsymbol{S} = (\boldsymbol{s}_1, \boldsymbol{s}_2, \ldots, \boldsymbol{s}_n)$, $\odot$ denotes the Hadamard product, $f$ is performed on each column of $\tilde{\boldsymbol{X}}$ separately, and $\mathcal{F}$ denotes a hypothesis set of functions. We have infinite choices for $\mathcal{F}$. For example, $\mathcal{F}$ can be a set of functions in the form of neural network with some parameters $W \in \mathcal{W}$, where $\mathcal{W}$ denotes a set of matrices under some constraints. In this case, problem (1) defines a denoising autoencoder or stacked denoising autoencoders (Vincent et al., 2010), where the noises are introduced by filling the missing ratings with zeros. Note that $f$ can be a general neural network and is not necessarily an encoder-decoder architecture.

Let $f$ be an autoencoder with linear activation function in the output layer. Then (1) becomes

$$\underset{\boldsymbol{W}_1, \boldsymbol{W}_2}{\text{minimize}} \left\| \boldsymbol{S} \odot (\tilde{\boldsymbol{X}} - \boldsymbol{W}_2 \sigma(\boldsymbol{W}_1 \tilde{\boldsymbol{X}})) \right\|_F^2 + \lambda \left( \|\boldsymbol{W}_1\|_F^2 + \|\boldsymbol{W}_2\|_F^2 \right), \tag{2}$$

where $\boldsymbol{W}_1 \in \mathbb{R}^{d \times m}$ and $\boldsymbol{W}_2 \in \mathbb{R}^{m \times d}$ are weights matrices to learn and $\lambda$ is a nonnegative constant to control the strength of regularization. We have omitted the bias terms for simplicity. $\sigma$ denotes an activation function such as ReLU $\sigma(x) = \max(x, 0)$ and Sigmoid $\sigma(x) = 1/(1 + \exp(-x))$.

Note that (2) is exactly the basic model considered by (Sedhain et al., 2015; Wu et al., 2016; Muller et al., 2018; Yi et al., 2020). Once (2) is used and $d$ is much smaller than $\min(m, n)$, the following assumption is made implicitly: $\tilde{\boldsymbol{X}}$ can be well approximated by the low-rank matrix $\boldsymbol{W}_2 \sigma(\boldsymbol{W}_1 \tilde{\boldsymbol{X}})$. However, this assumption does not always hold in real applications. Consider a data generating model $\boldsymbol{X} = h(\boldsymbol{A}' \boldsymbol{B}')$, where $h : \mathbb{R}^1 \to \mathbb{R}^1$ is an element-wise nonlinear function and $\boldsymbol{A}' \in \mathbb{R}^{m \times d}$, $\boldsymbol{B}' \in \mathbb{R}^{d \times n}$ may be generated by some nonlinear functions. If the nonlinearity of $h$ is high, $\boldsymbol{X}$ cannot be well approximated by a rank-$d$ matrix. This analysis indicates that if the element-wise nonlinearity in generating $\boldsymbol{X}$ is strong, (2) should use a large $d$ to ensure a small training error.

The element-wise nonlinearity widely exists in real data. For example, in imaging science, the intensity of pixels are nonlinear responses of photoelectric element to the spectrum. In chemical engineering, many sensors have nonlinear responses. In biomedical engineering, the dose-responses are often nonlinear curves. Hence, in collaborative filtering, the ratings may be nonlinear responses to some latent values, according to the studies on response curves in neuroscience and psychology (LeDoux, 2000; Baker, 2001). Therefore, instead of (2), one may consider the following problem

$$\underset{\boldsymbol{W}_1, \boldsymbol{W}_2}{\text{minimize}} \left\| \boldsymbol{S} \odot (\tilde{\boldsymbol{X}} - h(\boldsymbol{W}_2 \sigma(\boldsymbol{W}_1 \tilde{\boldsymbol{X}}))) \right\|_F^2 + \lambda \left( \|\boldsymbol{W}_1\|_F^2 + \|\boldsymbol{W}_2\|_F^2 \right), \tag{3}$$

where $h$ should be determined beforehand. A naive approach to determining $h$ is choosing a bounded or partially bounded nonlinear function according to the range of the data. For example, if the data are image pixels within $[0, 1]$, one may use Sigmoid. If the data are nonnegative, one may use ReLU. However, such choices only considered the range of the data, which is just a small portion of the nonlinearity. Within the range, the true response functions are not necessarily linear (ReLU) or related to exponential (Sigmoid), and can be much more complicated.

As it is difficult to determine $h$ in advance, we propose to learn $h$ from the data adaptively. We have different approaches to learning $h$. The first approach is combining various activation functions, i.e., $h_\theta(z) = \sum_{i=1}^{k} \theta_i \sigma_i(z)$, where $\sigma_i(\cdot)$ are different activation functions and $\boldsymbol{\theta} = (\theta_1, \ldots, \theta_k)^\top$ are parameters to estimate. However, it is not clear whether this is able to approximate a wide range of nonlinear functions. The second approach is using polynomial functions, i.e., $h_\theta(z) = \sum_{i=1}^{k} \theta z^k$. It is a $k$-order polynomial function and can well approximate any smooth functions provided that $k$

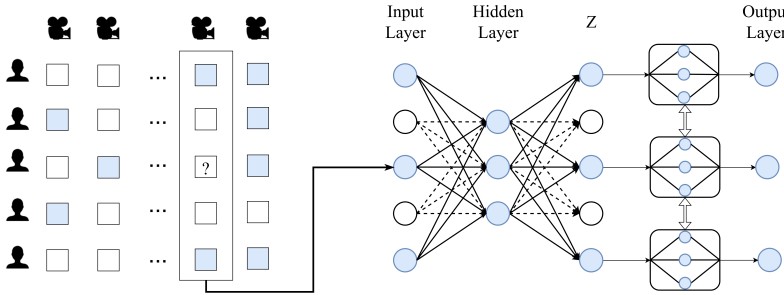

Figure 1: Example schematic of AEMC-NE with one hidden layer in each of the two networks. Users (variables) and items (samples) are represented by rows and columns of the matrix, in which observed ratings are colored and unobserved ratings are left white. Each target rating being predicted is marked with a question mark. The element-wise network is shared for all output nodes. The roles of users and items can be exchanged, namely, users could be samples and items could be variables.

is sufficiently large. Another approach is using a neural network, i.e.,

$$h_\Theta(z) = \boldsymbol{\Theta}_{L_\Theta}(\sigma_\Theta(\boldsymbol{\Theta}_{L_\Theta-1}\sigma_\Theta(\cdots\sigma_\Theta(\boldsymbol{\Theta}_1 z)\cdots))), \qquad (4)$$

where $\boldsymbol{\Theta}_1, \boldsymbol{\Theta}_{L_\Theta}$ are vectors, $\boldsymbol{\Theta}_2, \dots, \boldsymbol{\Theta}_{L_\Theta-1}$ are matrices, and $\sigma_\Theta$ is a fixed activation function. According to the universal approximation theorems (Pinkus, 1999; Sonoda & Murata, 2017; Lu et al., 2017), (4) can approximate any continuous functions when the network is wide or deep enough. As (4) is more flexible than other choices in function approximation, we propose to solve

$$\underset{W,\Theta}{\text{minimize}} \; \left\|\boldsymbol{S} \odot \left(\tilde{\boldsymbol{X}} - h_\Theta\big(g_W(\tilde{\boldsymbol{X}})\big)\right)\right\|_F^2 + \lambda_W \sum_{l=1}^{L_W} \|\boldsymbol{W}_l\|_F^2 + \lambda_\Theta \sum_{l=1}^{L_\Theta} \|\boldsymbol{\Theta}_l\|_F^2, \qquad (5)$$

where $W = \{\boldsymbol{W}_1, \dots, \boldsymbol{W}_{L_W}\}$, $\Theta = \{\boldsymbol{\Theta}_1, \dots, \boldsymbol{\Theta}_{L_\Theta}\}$, and

$$g_W(\tilde{\boldsymbol{X}}) = \boldsymbol{W}_{L_W}\big(\sigma_W\big(\boldsymbol{W}_{L_W-1}\sigma_W(\cdots\sigma_W(\boldsymbol{W}_1\tilde{\boldsymbol{X}})\cdots)\big)\big). \qquad (6)$$

In addition, we assume $\boldsymbol{W}_l \in \mathbb{R}^{d_l \times d_{l-1}}$, $l \in [L_W]$, and $\boldsymbol{\Theta}_l \in \mathbb{R}^{p_l \times p_{l-1}}$, $l \in [L_\Theta]$. Note that $d_0 = d_{L_W} = m$ and $p_0 = p_{L_\Theta} = 1$. Comparing (5) with (1), we see that we have replaced $f$ by $h_\Theta \circ g_W$ with Frobenius-norm constrained weight matrices. Model (5) is exactly our *neuron-enhanced autoencoder-based matrix completion* (AEMC-NE) method. There are two different neural networks. The first one is an autoencoder defined by $h_\Theta \circ g_W$, which is to learn a contraction map from the incomplete data matrix $\tilde{\boldsymbol{X}}$ to itself or its observed entries more precisely. The second neural network is performed in an element-wise manner to learn an activation function $h$ adaptively for the output layer of the autoencoder. Figure 1 shows an example schematic of AEMC-NE.

It is worth noting that, shown in Figure 1, the network architecture of the proposed method cannot be easily adapted for new users because we need to, at least, change the size of the output layer of the main neural network. In the output layer of the main network, we add one node for each new user and train the weights between the last but one layer and the new output nodes using the ratings of these new users, where we are not considering the cold-start problem. It is also worth noting that (5) does not make any assumption or take advantage of the prior knowledge about the missing mechanism of $\boldsymbol{X}$. When the mechanism is missing not at random, supposing the entries are observed with different probabilities (e.g., $p_{ij}$ for $X_{ij}$, $(i,j) \in S$), a better approach is to replace $\boldsymbol{S}$ with $\boldsymbol{Q}$, where $Q_{ij} = \hat{p}_{ij}^{-1/2}$ and $\hat{p}_{ij}$ denotes an estimation for $p_{ij}$.

**Optimization and complexity analysis** AEMC-NE (5) can be solved by various optimizers such as gradient descent and Adam (Kingma & Ba, 2015). We analyze the time and space complexity of AEMC-NE with a gradient-based optimizer. In each iteration (suppose the batch size is $b$), the time complexity is $O\big(b\sum_{l=1}^{L_W} d_l d_{l-1} + mb\sum_{l=1}^{L_\Theta} p_l p_{l-1}\big)$, in which the first part is from the main neural network and the second part is from the element-wise neural network. The space complexity is $O\big(b\sum_{l=0}^{L_W} d_l + \sum_{l=0}^{L_W} d_l d_{l-1} + mb\sum_{l=1}^{L_\Theta} p_l + \sum_{l=0}^{L_\Theta} p_l p_{l-1}\big)$. The element-wise neural network increased the time and space complexity to some extent. Considering the single hidden layer special case, the time and space complexity of AEMC-NE are $O\big(dmb + pmb\big)$ and $O\big(mb + pmb\big)$ respectively. Note that $p$ is usually much smaller than $d$, then the element-wise neural network just increased the time complexity slightly. Although the space complexity became $p$ times of the main network, it is still acceptable as $b \ll n$. See the time costs in Table 6.

## 3 THEORETICAL GUARANTEE

### 3.1 MISSING COMPLETELY AT RANDOM

In this section, we analyze the capability of AEMC-NE in predicting the missing values of $\boldsymbol{X}$. Note that $\frac{1}{|S|}\|\boldsymbol{S} \odot (\boldsymbol{X} - \hat{\boldsymbol{X}})\|_F^2 = \frac{1}{|S|}\|\boldsymbol{S} \odot (\tilde{\boldsymbol{X}} - \hat{\boldsymbol{X}})\|_F^2 \triangleq \mathcal{L}_S$, where $\hat{\boldsymbol{X}} = h_\Theta(g_W(\tilde{\boldsymbol{X}}))$. We have

$$\mathcal{L}_S = \frac{1}{|S|} \sum_{(i,j)\in S} \ell(X_{ij}, \hat{X}_{ij}),$$ where $\ell(X_{ij}, \hat{X}_{ij}) = (X_{ij} - \hat{X}_{ij})^2$. Note that instead of the square

loss, we may use other functions such as $|X_{ij} - \hat{X}_{ij}|$. In the remainder of this paper, $\ell(X_{ij}, \hat{X}_{ij})$ denotes a general loss. Let $S^c$ be the set of missing values of $\boldsymbol{X}$, i.e., $S^c = [m] \times [n]\backslash S$. The

generalization error is quantified by $\mathcal{L}_{S^c} = \frac{1}{|S^c|} \sum_{(i,j)\in S^c} \ell(X_{ij}, \hat{X}_{ij})$. We have[1].

**Theorem 3.1.** *Suppose a set $S$ of elements of $\boldsymbol{X} \in \mathbb{R}^{m\times n}$ are observed uniformly and randomly, which results in an incomplete matrix $\tilde{\boldsymbol{X}}$ with unknown ratings replaced by some values such as zero. Let $\hat{\boldsymbol{X}} = h_\Theta(g_W(\tilde{\boldsymbol{X}}))$, where $h_\Theta$ is defined by (4) and $g_W$ is defined by (6). Suppose $\|\boldsymbol{W}_l\|_2 \le a_l$, $\|\boldsymbol{W}_l\|_{2,1} \le a_l'$, $l \in [L_W]$, $\bar{d} := \max(d_1, \ldots, d_{L_{W-1}}) < m$, and $\|\boldsymbol{\Theta}_l\|_2 \le b_l$, $\|\boldsymbol{\Theta}_l\|_F \le b_l'$, $l \in [L_\Theta]$. Suppose the Lipschitz constants of $\sigma_W$ and $\sigma_\Theta$ are $\rho$ and $\varrho$ respectively. Suppose $\sup_{i,j} |\ell(X_{ij}, \hat{X}_{ij})| \le \tau_\ell$, $\ell$ is $\eta_\ell$-Lipschitz, and $\max(\|\tilde{\boldsymbol{X}}\|_\infty, \|\hat{\boldsymbol{X}}\|_\infty) \le \mu$. Then with probability at least $1 - \delta$ over the random sampling $S$,*

$$\mathcal{L}_{S^c} - \mathcal{L}_S \le \frac{C_1 \eta_\ell v_1 mn \ln(mn)}{|S||S^c|} + \frac{C_2 \eta_\ell \mu (mn)^{3/2} \sqrt{v_2 \ln v_3}}{|S||S^c|}$$
$$+ \frac{11\tau_\ell mn \sqrt{\min(|S|, |S^c|)}}{|S||S^c|} + 3\tau_\ell \sqrt{\frac{mn}{|S||S^c|} \ln\frac{1}{\delta}}, \tag{7}$$

*where $v_1 = \phi\sqrt{\ln m}\left(\sum_{l=1}^{L_W} \left(\frac{a_l'}{a_l}\right)^{2/3}\right)^{3/2}$, $v_2 = \sum_{l=1}^{L_\Theta} p_l p_{l-1}$, $v_3 = \phi L_\Theta \mu^{-1} \max_l \frac{b_l'}{b_l} > 1$, $\phi = \rho^{L_W - 1} \varrho^{L_\Theta - 1} \|\tilde{\boldsymbol{X}}\|_F \left(\prod_{l=1}^{L_W} a_l\right)\left(\prod_{l=1}^{L_\Theta} b_l\right)$, and $C_1$, $C_2$ are some absolute constants.*

First, let's show that the bound is non-trivial. Since activation functions are often at most 1-Lipschitz, we let $\rho = \varrho = 1$. Suppose $a_1 = \cdots = a_{L_W} = 1$ and $b_1 = \cdots = b_{L_\Theta} = 1$. Since $a_l'/a_l \le d_{l-1}$, we have $\left(\sum_{l=1}^{L_W} \left(a_l' a_l^{-1}\right)^{2/3}\right)^{3/2} \le L_W^{3/2} \max_l a_l' a_l^{-1} \le L_W^{3/2} \bar{d}$. In addition, $\max_l b_l'/b_l \le \max_l \sqrt{p_l}$. For convenience, hereinafter, we assume that $|S^c| \ge |S|$. Then the bound in Theorem 3.1 becomes

$$\mathcal{L}_{S^c} \le \mathcal{L}_S + \tilde{O}\left(\frac{\eta_\ell mn L_W^{3/2} \bar{d} \|\tilde{\boldsymbol{X}}\|_F}{|S||S^c|}\right) + \tilde{O}\left(\frac{\eta_\ell \mu (mn)^{3/2} \sqrt{\sum_{l=1}^{L_\Theta} p_l p_{l-1}}}{|S||S^c|}\right) + \Delta_\delta, \tag{8}$$

where $\Delta_\delta = \frac{11\tau_\ell mn}{|S^c|\sqrt{|S|}} + 3\tau_\ell \sqrt{\frac{mn}{|S^c||S|} \ln\frac{1}{\delta}}$. Note that $\|\tilde{\boldsymbol{X}}\|_F \le \sqrt{mn} \max_{ij} |\hat{X}_{ij}|$. If $|S||S^c| > C_3 \max\left(L_W^{3/2} \bar{d}(mn)^{3/2} \max_{ij} |\hat{X}_{ij}|, (mn)^{3/2} \sqrt{\sum_{l=1}^{L_\Theta} p_l p_{l-1}}\right)$ holds for some constant $C_3$, the bound is non-trivial. Obviously, the condition holds if $d$ is not too large and $|S|$ is close to $|S^c|$.

Particularly, Theorem 3.1 provides the following results.

**A. The element-wise network is helpful.**

Let $\mathcal{L}_S^0 = \frac{1}{|S|}\|\boldsymbol{S} \odot (\tilde{\boldsymbol{X}} - g_W(\tilde{\boldsymbol{X}}))\|_F^2$ and $\mathcal{L}_S = \frac{1}{|S|}\|\boldsymbol{S} \odot (\tilde{\boldsymbol{X}} - h_\Theta(g_W(\tilde{\boldsymbol{X}})))\|_F^2$ be the training errors of AEMC (AEMC-NE with the element-wise network ablated) and AEMC-NE respectively. $h_\Theta(\boldsymbol{z})$ can be reformulated as $h_\Theta(\boldsymbol{z}) = \boldsymbol{z} + r_{\bar{\Theta}}(\boldsymbol{z})$, where $r_{\bar{\Theta}}$ is a suitable neural network. Then $\mathcal{L}_S = \frac{1}{|S|}\|\boldsymbol{S} \odot (\tilde{\boldsymbol{X}} - g_W(\tilde{\boldsymbol{X}}) - r_{\bar{\Theta}}(g_W(\tilde{\boldsymbol{X}})))\|_F^2$. Thus, we always have $\Delta_{TE} := \mathcal{L}_S - \mathcal{L}_S^0 \le 0$ theoretically, though it is difficult to quantify $\Delta_{TE}$ because it depends on unknown data generating model. According to Theorem 3.1, the increase of complexity, denoted by $\Delta_{cpl}$, introduced by the element-wise network of AEMC-NE depends on $\varrho^{L_\Theta - 1}$, $\prod_{l=1}^{L_\Theta} b_l$, and $\frac{C_2 \eta_\ell \mu (mn)^{3/2} \sqrt{v_2 \ln v_3}}{|S||S^c|}$, which

---

[1] Given a matrix $\boldsymbol{Z}$, we use $\|\boldsymbol{Z}\|_2$, $\|\boldsymbol{Z}\|_{2,1} := \sum_i \|\boldsymbol{z}_i\|$, and $\|\boldsymbol{Z}\|_\infty := \max_{ij} |Z_{ij}|$ to denote the spectral norm, $\ell_{21}$ norm, and $\ell_\infty$ norm respectively.

are small. Suppose the nonlinearity of the response function existing in $\mathbf{X}$ is strong enough, we have $\Delta_{TE} + \Delta_{cpl} < 0$. It means AEMC-NE guarantees a tighter upper bound of test error than AEMC.

On the other hand, suppose we add more ($L_\Theta$, to be more precise) layers to the decoder of AEMC and obtain an AEMC+ that has the same expressive ability as AEMC-NE. Then AEMC+ and AEMC-NE have the same training error and we only need to compare the model complexity. As explained in Appendix B, compared with AEMC-NE, AEMC+ has $2m(m-1)p_1 + (L_\Theta - 2)m(m-1)p^2 \triangleq \Delta_{par}$ more parameters, where $p = \min_{1 \le l \le L_\Theta - 1} p_l$. Obviously, $\Delta_{par}$ is a large number. More formally, for AEMC+, in Theorem 3.1, $\phi$ becomes $\phi' = \rho^{L_W - 1} \rho^{L_\Theta - 1} \|\tilde{\mathbf{X}}\|_F \left( \prod_{l=1}^{L_W} a_l \right) \left( \prod_{l=L_W+1}^{L_W+L_\Theta} a_l \right)$ and $v_1$ becomes $v_1' = \phi' \sqrt{\ln m} \left( \sum_{l=1}^{L_W+L_\Theta} \left( \frac{a_l'}{a_l} \right)^{2/3} \right)^{3/2}$. Since the element-wise network (width $p$) in AEMC-NE is much smaller than the added layers (width $pm$, see Figure 3) in AEMC+, we have $v_1' > v_1$, which means AEMC+ has a looser generalization error bound than AEMC-NE.

**B. Filling the missing values with zeros is helpful.**
Suppose we have two methods to fill the unknown ratings, one is zero filling and the other is a method different from zero filling (e.g. random filling or mean filling). Suppose based on the two methods, we get the same training errors, i.e., $\mathcal{L}_S^{\text{zero}} = \mathcal{L}_S^{\text{other}}$. Using Theorem 3.1 and the fact $\|\tilde{\mathbf{X}}^{\text{zero}}\|_F < \|\tilde{\mathbf{X}}^{\text{other}}\|_F$, we conclude that the upper bound of $\mathcal{L}_{S^c}^{\text{zero}}$ is less than the upper bound of $\mathcal{L}_{S^c}^{\text{other}}$. This result has been empirically verified by many previous work (e.g., (Sedhain et al., 2015; Muller et al., 2018; Yi et al., 2020)) in which the missing values are filled by zeros rather than the means or other values. It is worth noting that although the zero-filling, compared with some other weak imputation methods, may lead to a larger training error $\mathcal{L}_S^{\text{zero}}$, the $v_1$ in equation 7 can be much smaller, which finally leads to a tighter bound of $\mathcal{L}_{S^c}^{\text{zero}}$. However, if we have a perfect imputation (e.g., filling the missing values with the true values), the training error $\mathcal{L}_S^{\text{other}}$ can be very small, though the corresponding $v_1$ is larger, which eventually leads to a tight bound of $\mathcal{L}_{S^c}^{\text{other}}$.

**C. Increasing $n$ reduces the upper bound.**

Let the sampling rate $\frac{|S|}{mn} \triangleq \zeta$ and network structures be fixed. Then (8) becomes

$$\mathcal{L}_{S^c} \le \mathcal{L}_S + \tilde{O}\left( \frac{\eta_\ell L_W^{3/2} \max_{ij} |\tilde{X}_{ij}|}{\zeta(1-\zeta)mn/d} \right) + \tilde{O}\left( \frac{\eta_\ell \mu \sqrt{\sum_{l=1}^{L_\Theta} p_l p_{l-1}}}{\zeta(1-\zeta)\sqrt{mn}} \right) + \frac{11\tau_\ell}{\sqrt{(1-\zeta)\zeta^2 mn}} + 3\tau_\ell \sqrt{\frac{1}{(1-\zeta)\zeta mn} \ln \frac{1}{\delta}}.$$

Therefore, when $n$ increases, the bound becomes tighter. In addition, $\mathcal{L}_{S^c} \le \mathcal{L}_S$ when $n \to \infty$. In real applications, if there are more users than items, we construct a neural network such that the input is a vector of each user's rating, where items correspond to features and users correspond to samples. In other words, a larger difference between the number of users and the number of items (or the number of variables and the number of samples more generally) leads to a tighter upper bound, because we can construct the autoencoder along the smaller size of the matrix.

## 3.2 MISSING NOT AT RANDOM

It should be pointed out that Theorem 3.1 is based on the assumption of missing completely at random (MCAR). The MCAR assumption has been widely used in previous work of matrix completion (Srebro & Shraibman, 2005; Candès & Recht, 2009; Shamir & Shalev-Shwartz, 2014; Fan et al., 2019) and missing data imputation (Yoon et al., 2018; Mattei & Frellsen, 2019). However, it has been found that in collaborative filtering, the data entries may be missing not at random (MNAR) (Marlin et al., 2007; Marlin & Zemel, 2009; Schnabel et al., 2016; Wang et al., 2019; Ma & Chen, 2019). Hence, in addition to Theorem 3.1, we propose a generalization error bound under the assumption of MNAR, in which the entries of the matrix are observed with different probabilities.

**Theorem 3.2.** *Suppose a set $S$ of elements of $\mathbf{X} \in \mathbb{R}^{m \times n}$ are observed, where $X_{ij}$ is observed with probability $p_{ij}$, $\forall (i,j) \in S$. Let $\hat{\mathbf{X}} = h_\Theta(g_W(\tilde{\mathbf{X}}))$. Based on the same definitions of $\tilde{\mathbf{X}}$, $h_\Theta$, $g_W$, $\tau_\ell$, $\eta_\ell$, $v_1$, $v_2$, $v_3$ used in Theorem 3.1, with probability at least $1 - \frac{2}{mn}$,*

$$\frac{1}{mn} \sum_{(i,j) \in [m] \times [n]} \ell(X_{ij}, \hat{X}_{ij}) - \frac{1}{mn} \sum_{(i,j) \in S} p_{ij}^{-1} \ell(X_{ij}, \hat{X}_{ij})$$

$$\le \frac{\sqrt{\sum_{(i,j) \in S} p_{ij}^{-2}}}{mn} \left( C_1' \eta_\ell v_1 + C_2' \tau_\ell \sqrt{v_2 \ln v_3} \right) + \tau_\ell \left( \frac{1}{\sqrt{mn}} + \frac{\sqrt{\sum_{(i,j) \in S} p_{ij}^{-2}}}{mn} \right), \tag{9}$$

*where $C_1'$ and $C_2'$ are some absolute constants.*

In the theorem, $\sqrt{\sum_{(i,j)\in S} p_{ij}^{-2}}/(mn)$ is at the scale of $1/\sqrt{|S|}$, provided that the variance of $p_{ij}$ is not too large. The RHS of the bound is comparable to that in Theorem 3.2. Therefore, the conclusions A, B, C for Theorem 3.1 also hold for Theorem 3.2. In reality, $p_{ij}$ are unknown but can be estimated via some approaches (Schnabel et al., 2016; Wang et al., 2019; Ma & Chen, 2019). Based the estimation (denoted by $\hat{p}_{ij}$), we can obtain a bound for $\frac{1}{mn} \sum_{(i,j)\in[m]\times[n]} \ell(X_{ij}, \hat{X}_{ij}) - \frac{1}{mn} \sum_{(i,j)\in S} \hat{p}_{ij}^{-1} \ell(X_{ij}, \hat{X}_{ij})$, which is shown by Theorem A.1 in the supplementary material.

## 4 CONNECTION WITH PREVIOUS WORK

The element-wise neural network of AEMC-NE can be regarded as an activation function adaptively learned from data. It is related to the previous work on adaptive activation functions such (Lin et al., 2013; Agostinelli et al., 2014; Hou et al., 2017; Goyal et al., 2019). For instance, Hou et al. (2017) showed that applying adaptive activation functions in the regression (second-to-last) layer of a neural network can significantly decrease the bias. Their adaptive activation function is in the form of piece-wise polynomials. We found that, empirically, in AEMC-NE, the improvement given by polynomials (Hou et al., 2017) is not significant, possibly due to the unboundedness of polynomials.

The theoretical study for autoencoder and deep learning based collaborative filtering is very limited. Recently a few researchers studied the generalization ability or sample complexity of deep neural networks (Bartlett et al., 2017; Neyshabur et al., 2018; Golowich et al., 2018) but their results do not apply to autoencoder-based CF or missing data imputation as well as our AEMC-NE.

Shamir & Shalev-Shwartz (2014) provided the following generalization bound for nuclear norm minimization based CF: $\mathcal{L}_{S^c} \leq \mathcal{L}_S + O\left(\frac{\eta_\ell \|\hat{X}\|_*(\sqrt{m}+\sqrt{n})}{|S|}\right) + R$, where $\hat{X}$ denotes the recovered matrix given by nuclear norm minimization and $R$ stands for the remainder of their result (Theorem 6). Suppose the rank of $\hat{X}$ is $\bar{d}$. Then the term related to the nuclear norm can be as large as $O\left(\frac{\eta_\ell \sqrt{\bar{d}}\|\hat{X}\|_F(\sqrt{m}+\sqrt{n})}{\zeta mn}\right)$ or $O\left(\eta_\ell \mu \sqrt{\frac{\bar{d}}{\zeta^2 m}}\right)$, where we have assumed $m < n$. According to the result C of Theorem 3.1, the dominating term in our bound can be $\tilde{O}\left(\eta_\ell \mu \sqrt{\frac{\bar{d}}{\zeta^2 m}} \sqrt{\frac{L_W^3 \zeta \bar{d}}{n}}\right)$. Note that with the same $\bar{d}$, the training error of our AEMC-NE is less than the training error of nuclear norm minimization because we are using neural networks. Now we conclude that when $n$ is sufficiently large (compared to $L_W^3 \zeta \bar{d}$), our bound is tighter than that of (Shamir & Shalev-Shwartz, 2014).

## 5 NUMERICAL RESULTS

### 5.1 EXPERIMENTS ON SYNTHETIC DATA

We generate a synthetic data matrix $\mathbf{X}$ of size $300 \times 3000$ using a latent (10 dimension) variable model with a nonlinear response function (detailed in Appendix C). We randomly drop a certain portion of the elements of $\mathbf{X}$ and aim to recover the missing values. This setting is MCAR. We compare our AEMC-NE with AEMC as well as its variants under the evaluation of the following metric *relative recovery error* $= \|(\mathbf{X} - \hat{\mathbf{X}}) \odot \mathbf{S}\|_F / \|\mathbf{X} \odot \mathbf{S}\|_F$. The results (average of 10 runs) are reported in Figure 2. We see that, in Figure 2(a), AEMC-NE outperforms AEMC and its variants (with more hidden layers) in all cases when the missing rate increases from 0.1 to 0.8 and the superiority of AEMC-NE is significant when the missing rate is not too high. Importantly, this result is consistent with the conclusion A given by Theorem 3.1. Figure 2(b) shows that AEMC-NE is not sensitive to the size of the element-wise network provided that it is not too small. Figure 2(c) shows that both AEMC and AEMC-NE are not sensitive to the width of the middle layer of the autoencoder. Figure 2(d) indicates that relatively small weight decays have little impact.
Besides MCAR, we also conduct some experiments of MNAR. The relative recovery errors (over five runs) are reported in Table 1, where the missing probability varies along the locations of the entries in the matrix. The detailed setting is in Appendix C. We see that our AEMC-NE outperformed the baseline AEMC significantly in all cases. We can compare this table (MNAR) with the results in Figure 2 (MCAR) and find that the performances are similar. These results verify that our method

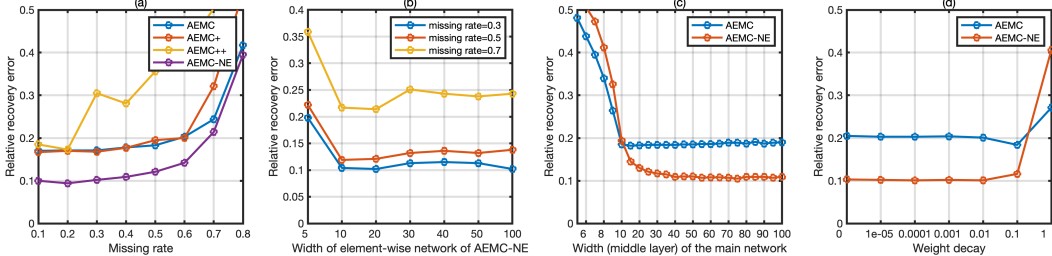

Figure 2: Recovery performance on synthetic data in the setting of **MCAR**. **(a)** Influence of missing rate. Network structure: AEMC 300-100-30-100-300; AEMC+ 300-100-30-100-100-300; AEMC++ 300-100-30-100-100-100-300; AEMC-NE 300-100-30-100-300 for the main network and $1 - w - w - 1$ for the element-wise network, where $w = 20$. **(b)** AEMC-NE with different width ($w$) of the hidden layers in the element-wise network. **(c)** Influence of the width (middle layer) of the main network. The missing rate is $0.5$ and $w = 20$. **(d)** Influence of the weight decay ($\lambda_W = \lambda_\Theta$). The missing rate is $0.5$ and $w = 20$.

works well for both MCAR and MNAR, AEMC-NE outperforms AEMC, and the element-wise network of AEMC-NE is indeed effective in improving recovery accuracy.

Table 1: Rcovery performance on synthetic data in the setting of **MNAR**

| Missing rate | 0.2 | 0.3 | 0.4 | 0.5 | 0.6 | 0.7 |
|---|---|---|---|---|---|---|
| AEMC | $0.170_{\pm 0.002}$ | $0.171_{\pm 0.001}$ | $0.180_{\pm 0.003}$ | $0.187_{\pm 0.002}$ | $0.211_{\pm 0.003}$ | $0.268_{\pm 0.004}$ |
| AEMC-NE | $0.101_{\pm 0.002}$ | $0.106_{\pm 0.003}$ | $0.109_{\pm 0.004}$ | $0.124_{\pm 0.004}$ | $0.153_{\pm 0.005}$ | $0.243_{\pm 0.006}$ |

## 5.2 EXPERIMENTS ON MOVIELENS DATASETS

In this section, we evaluate the proposed method AEMC-NE on Movielens-100K, Movielens-1M, and Movielens-10M (Harper & Konstan, 2015). These datasets contain real-world ratings for 1682, 3900, and 10000 movies given by 943, 6040, and 72000 users respectively. We randomly sample 90% of the known ratings as training set, leaving the remaining 10% as the test set. Among the training set, 5% are held out for hyperparameter tuning. The model performance is evaluated by the root mean squared error defined as RMSE $= \sqrt{\frac{1}{|S^c|}\sum_{(i,j)\in S^c}(X_{ij} - \hat{X}_{ij})^2}$, where $S^c$ denotes the set of test ratings. In our AEMC-NE, the main neural network has one hidden layer. The numbers of hidden units are 500, 900, and 1000 for MovieLens-100k, MovieLens-1M, and MovieLens-10M respectively. The element-wise neural network has one hidden layer, of which the size is 200. The regularization parameters $\lambda_W$ and $\lambda_\Theta$ were chosen from $\{0.01, 0.1, 1, 10, 50, 100, 200, 500\}$. The optimizer for AEMC-NE is Adam (Kingma & Ba, 2015).

In Table 2, we report the mean RMSE of our AEMC-NE in comparison to a few baselines based on 20 random splits. Our AEMC-NE outperformed all baselines on MovieLens-100k and MovieLens-1M. We have tried to increase the depth of the main neural network and the element-wise neural network, but the improvements in terms of RMSE are not significant.

## 5.3 EXPERIMENTS ON DOUBAN AND FLIXSTER

We use the preprocessed subsets and splits provided by Monti et al. (2017). The datasets both contain 3000 users and 3000 items. Douban contains 136,891 ratings with density 0.0152 on a rating scale $\{1, 2, 3, 4, 5\}$. Flixster contains 26,173 ratings with density 0.0029 on a rating scale $\{0.5, 1, 1.5, ..., 5\}$. Five percent of the training samples are used for hyperparameter tuning. In AEMC-NE, the main network has one hidden layer, of which the size is 500, for both datasets. The structure of the element-wise network is the same as that used for the MovieLens datasets. In Table 3, we show the mean RMSE on 5 repeated experiments. AEMC-NE outperforms other baselines on both datasets. Note that some compared methods include extra content like the side information of users and items into the model, while AEMC-NE does not require extra content.

Table 2: RMSE results on the three MovieLens datasets

| Model | ML-100k | ML-1M | ML-10M |
|---|---|---|---|
| BiasMF (Koren et al., 2009) | 0.911 | 0.845 | 0.803 |
| NNMF (Dziugaite & Roy, 2015) | 0.903 | 0.843 | - |
| LLORMA (Lee et al., 2016) | 0.8881 | 0.833 | 0.782 |
| GC-MC (van den Berg et al., 2017) | 0.905 | 0.832 | 0.777 |
| AutoSVD++ (Zhang et al., 2017) | 0.904 | 0.848 | - |
| AutoSVD (Zhang et al., 2017) | 0.901 | 0.86 | - |
| CF-NADE (Zheng et al., 2016) | - | 0.829 | 0.771 |
| DMF+ (Yi et al., 2019) | 0.8889 | 0.8321 | - |
| IMC-GAE (Shen et al., 2021) | 0.897 | 0.829 | - |
| GHRS (Zamanzadeh Darban & Valipour, 2022) | 0.8887 | 0.833 | 0.782 |
| AEMC (AutoRec) (Sedhain et al., 2015) | - | 0.831$\pm$ 0.003 | 0.782$\pm$ 0.003 |
| AEMC (AutoRec, reproduced) | 0.8818 $\pm$ 0.0082 | 0.8291$\pm$ 0.0021 | 0.7780$\pm$ 0.0024 |
| AEMC (ReLU output) | 0.8807 $\pm$ 0.0092 | 0.8276$\pm$ 0.0025 | 0.7851 $\pm$ 0.0029 |
| **AEMC-NE** (ours) | **0.8767** $\pm$ 0.0089 | **0.8248** $\pm$ 0.0024 | **0.7723** $\pm$ 0.0025 |

Table 3: RMSE result of AEMC-NE and compared methods on Douban and Flixster dataset

| Model | Douban | Flixster |
|---|---|---|
| PMF (Mnih & Salakhutdinov, 2008) | 0.7492 | 0.9809 |
| GRALS (Rao et al., 2015) | 0.8326 | 1.245 |
| sRGCNN (Monti et al., 2017) | 0.801 | 0.926 |
| GC-MC (van den Berg et al., 2017) | 0.734 | 0.917 |
| Factorized EAE (Hartford et al., 2018) | 0.738 | 0.908 |
| GRAEM (Strahl et al., 2020) | 0.7497 | 0.8857 |
| AEMC (AutoRec, reproduced) | 0.7325$\pm$ 0.0009 | 0.957$\pm$ 0.0003 |
| AEMC (ReLU output) | 0.7306 $\pm$ 0.0010 | 0.9610 $\pm$ 0.0005 |
| **AEMC-NE** (ours) | **0.7286**$\pm$ 0.0007 | **0.8816**$\pm$ 0.0003 |

## 5.4 ADDITIONAL ANALYSIS FOR CF EXPERIMENTS

Note that in Tables 2 and 3, the improvement given by AEMC-NE is not very significant compared with the linear models such as LLORMA. The reason is that the rating matrices of these datasets are square or nearly square. Our theoretical results have shown that the superiority of AEMC-NE is more significant when the rating matrix is very fat or tall. Here we consider a subset of MovieLens-1M consisting of only 500 users who rated most. Then the size of the rating matrix is $3706 \times 500$ and the train-test ratio is $9 : 1$. The RMSEs (average of 10 runs) of SVD, SVD++, AEMC, AEMC$^+$, AEMC$^{++}$ (one or two more hidden layers compared to AEMC) and AEMC-NE are reported in Table 4. AEMC-NE outperformed the three baselines largely, especially when compared to the results in Tables 2 and 3. This result is consistent with our theoretical analysis.

Table 4: RMSEs on MovieLens-1M subset

| Method | SVD | SVD++ | AEMC | AEMC$^+$ | AEMC$^{++}$ | AEMC-NE | AEMC$^+$-NE |
|---|---|---|---|---|---|---|---|
| RMSE | 0.8713 | 0.8591 | 0.8265 | 0.8294 | 0.8308 | **0.8182** | **0.8207** |

**More results** The time cost comparison, results of NDCG metric, experiments of CF in the case of MNAR, and missing data imputation on UCI datasets are in Appendices E, F, and G respectively.

## 6 CONCLUSION

This paper presented a novel collaborative filtering method called AEMC-NE, which is composed of two neural networks: one is a layer-wise network and the other is an element-wise network that is able to learn an activation function for the output layer adaptively. We also analyzed the generalization error bounds for AEMC-NE, which verified the effectiveness of AEMC-NE theoretically. AEMC-NE outperformed many baselines on a few benchmark datasets of collaborative filtering. Note that our method can be extended to the scenarios of implicit feedback if we use negative sampling and ranking loss functions such as the BPR loss (Rendle et al., 2009). In addition, AEMC-NE is applicable to more general missing data imputation problems.

ACKNOWLEDGEMENTS

This work was supported by the National Natural Science Foundation of China under Grants No.62106211 and No.62072151, the research funding T00120210002 of Shenzhen Research Institute of Big Data, the funding UDF01001770 of The Chinese University of Hong Kong, Shenzhen, Anhui Provincial Natural Science Fund for the Distinguished Young Scholars (2008085J30), Open Foundation of Yunnan Key Laboratory of Software Engineering (2023SE103), CCF-Baidu Open Fund, and CAAI-Huawei MindSpore Open Fund.

ETHICS STATEMENTS

The authors of this work have read and commit to adhering to the ICLR Code of Ethics.

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

## A  MORE THEORETICAL RESULT

The following theorem provides a generalization error bound for missing data imputation when the missing pattern is missing not at random and the propensities are estimated.

**Theorem A.1.** *Suppose a set $S$ of elements of $\boldsymbol{X} \in \mathbb{R}^{m \times n}$ are observed, where $X_{ij}$ is observed with probability $p_{ij}$, $\forall (i,j) \in S$. Suppose the estimated propensities are $\hat{p}_{ij}$, $\forall (i,j) \in S$. Let $\hat{\boldsymbol{X}} = h_\Theta(g_W(\tilde{\boldsymbol{X}}))$. Based on the same conditions or definitions of $\tilde{\boldsymbol{X}}$, $h_\Theta$, $g_W$, $\tau_\ell$, $\eta_\ell$, $v_1$, $v_2$, $v_3$ used in Theorem 3.1, with probability at least $1 - \frac{2}{mn}$,*

$$
\frac{1}{mn} \sum_{(i,j) \in [m] \times [n]} \ell(X_{ij}, \hat{X}_{ij}) - \frac{1}{mn} \sum_{(i,j) \in S} \hat{p}_{ij}^{-1} \ell(X_{ij}, \hat{X}_{ij})
$$

$$
\leq \frac{\sqrt{\sum_{(i,j) \in S} \hat{p}_{ij}^{-2}}}{mn} \left( C_1' \eta_\ell v_1 + C_2' \tau_\ell \sqrt{v_2 \ln v_3} \right) + \tau_\ell \left( \frac{1}{\sqrt{mn}} + \frac{\sqrt{\sum_{(i,j) \in S} \hat{p}_{ij}^{-2}}}{mn} \right) \tag{10}
$$

$$
+ \frac{\tau_\ell}{mn} \sum_{(i,j) \in [m] \times [n]} \left| 1 - \frac{p_{ij}}{\hat{p}_{ij}} \right|,
$$

*where $C_1'$ and $C_2'$ are some absolute constants.*

## B CONNECTION BETWEEN THE NETWORK STRUCTURES OF AEMC AND AEMC-NE

Figure 3 shows an example of the network architecture of AEMC-NE. We see that adding the orange connections forms a new neural network without an element-wise network. The new network can be regarded as an extension of AEMC (denoted by AEMC+), of which the decoder has more layers. It indicates that adding more layers to the decoder of AEMC and ensuring sparse and shared weights yield AEMC-NE.

In Figure 3, intuitively, the number of orange connections is much larger than the number of black connections. To be more precise and general, suppose the element-wise network of AEMC-NE has $k$ hidden layers of width $p$ and the decoder of AEMC+ has $k$ additional hidden layers, then AEMC+ has $2m(m-1)p + (k-1)m(m-1)p^2$ more parameters than AEMC-NE. Compared with AEMC, AEMC+ introduces an additional multiplication $\rho^{k+1}\prod_{l=L_W+1}^{L_W+k+1} a_l$ (which could be much larger than one) to the $\phi$ in Theorem 3.1. Thus, AEMC+ has a higher generalization error bound than AEMC-NE.

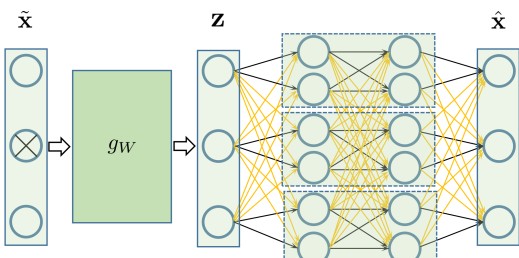

Figure 3: Network structure of AEMC-NE or AEMC (with more layers in the decoder)

## C EXPERIMENTAL SETTING ABOUT THE SYNTHETIC DATA

We generate synthetic data using the following nonlinear latent variable model

$$\boldsymbol{X} = \boldsymbol{W}_3 Tanh(\boldsymbol{W}_2 Tanh(\boldsymbol{W}_1 Z)),$$

$$\boldsymbol{X} = (Cosine(\boldsymbol{X}) + \boldsymbol{X}),$$

where $\boldsymbol{W}_1 \in \mathbb{R}^{50\times10}, \boldsymbol{W}_2 \in \mathbb{R}^{100\times50}, \boldsymbol{W}_3 \in \mathbb{R}^{300\times100}$ are drawn from the uniform distribution of $[-1, 1]$, and $\boldsymbol{Z} \in \mathbb{R}^{10\times3000}$ is random Gaussian matrix.

For MNAR, we partition the data matrix into four blocks, i.e., $\boldsymbol{X} = \begin{bmatrix} \boldsymbol{X}_1 & \boldsymbol{X}_2 \\ \boldsymbol{X}_3 & \boldsymbol{X}_4 \end{bmatrix}$. In block $\boldsymbol{X}_i$, the elements are missing with probability $p_i, i = 1, 2, 3, 4$. We let $p_1 < p_2 < p_3 < p_4$. The correspondence between the total missing rate and $p_i$ is in Table 5. We see that the differences between the missing rates are large. For instance, when the total missing rate is $0.2, p_4/p_1 = 21$.

Table 5: Details of MNAR setting on the synthetic data

| total missing rate | 0.2 | 0.3 | 0.4 | 0.5 | 0.6 | 0.7 |
|---|---|---|---|---|---|---|
| $p_1, p_2$ | 0.02, 0.13 | 0.12, 0.22 | 0.23, 0.33 | 0.33, 0.42 | 0.42, 0.53 | 0.53, 0.63 |
| $p_3, p_4$ | 0.23, 0.43 | 0.33, 0.53 | 0.43, 0.62 | 0.52, 0.72 | 0.63, 0.83 | 0.73, 0.93 |
| $p_4/p_1$ | 21.5 | 4.42 | 2.70 | 2.28 | 1.98 | 1.75 |

## D THE INFLUENCE OF HIDDEN UNITS NUMBER IN AEMC-NE

Figure 4 shows the influence of the number of hidden units in each of the neural networks of NE-AEMC on MovieLens-1M. In the left plot, the number of hidden units of the element-wise neural

network is fixed as 200. In the right plot, the number of hidden units of the main neural network is fixed as 900. We see that in NE-AEMC, the main neural network with 900 hidden units and the element-wise neural network with 200 hidden units provide the best performance.

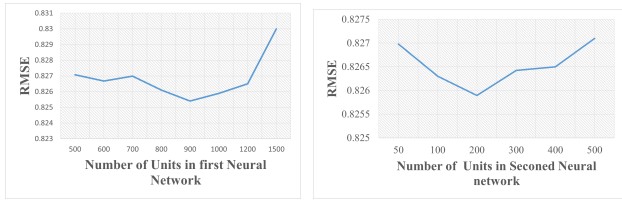

Figure 4: Influence of the number of hidden units in each of the two neural networks of NE-AEMC on MovieLens-1M.

# E  TIME COSTS COMPARISON AND NDCG RESULTS

We compare the time costs of AEMC-CF and AEMC (Sedhain et al., 2015) in Table 6. On each of the five datasets, the time cost of our NE-AEMC is sligtly higher than that of AEMC, which is consistent with the theoretical time complexity we analyzed at the end of Section 2.

Table 6: Running time (second) of AEMC and NE-AEMC

| Dataset | AEMC | AEMC-NE |
|---------|------|---------|
| Douban | 12.1 | 17.8 |
| Flixster | 10.0 | 15.1 |
| ML-100k | 1.2 | 2.5 |
| ML-1M | 11.1 | 21.0 |
| ML-10M | 1820 | 2135 |

Besides RMSE results reported in the main paper, here we show the results in terms of the Normalized Discounted Cumulative Gain (NDCG) accumulated at a particular rank position $k$, which is defined as

$$\text{NDCG@k} = \frac{\text{DCG}_k}{\text{IDCG}_k}, \tag{11}$$

where $\text{DCG}_k = \sum_{i=1}^{k} \frac{\text{rel}_i}{\log_2(i+1)}$, $\text{IDCG}_k$ is the ideal discounted cumulative gain, and $\text{rel}_i$ is the graded relevance of the result at position $i$. It is obvious that

$$\text{RMSE} = 0 \implies \text{NDCG@}k = 1 \ \forall k. \tag{12}$$

This implies that a good RMSE often corresponds to a good NDCG. In recommendation systems, one may concern more about the top $k$ recommendations. That's why here we consider NDCG, in addition to RMSE.

The results of NDCG@$k$ ($k = 1, 5, 10, 50$) are reported in Table 7. We see that the proposed AEMC-NE outperformed AEMC in all cases. The improvement given by AEMC-NE is more significant when $k$ is smaller.

Table 7: Results of NDCG@10 and NDCG@50 on ML-100k and ML-1M.

| | | NDCG@1 | NDCG@5 | NDCG@10 | NDCG@50 |
|---|---|---|---|---|---|
| ML-100k | AEMC | 0.9059±0.0041 | 0.9313±0.0020 | 0.9451±0.0013 | 0.9678±0.0010 |
| | AEMC-NE | 0.9106±0.0054 | 0.9328±0.0025 | 0.9462±0.0014 | 0.9686±0.0009 |
| ML-1M | AEMC | 0.9197±0.0019 | 0.9325±0.0009 | 0.9441±0.0006 | 0.9678±0.0004 |
| | AEMC-NE | 0.9222±0.0013 | 0.9339±0.0007 | 0.9446±0.0006 | 0.9681±0.0003 |

It should be pointed out that in this paper, we are considering explicit feedback (ratings range from 1 to 5 or 10) rather than implicit feedback (binary feedback 0 or 1). So our results of NDCG cannot be compared with the results reported by many previous papers.

## F    EXPERIMENTS ABOUT MISSING NOT AT RANDOM

We also test AEMC (AutoRec) and AEMC-NE in the case of MNAR in comparison to MCAR. For MCAR, we follow the definition such that each element has 10% probability of being missing. For MNAR, we consider that higher ratings are more likely to be missing. The missing probability is set as $p(r) = Sigmoid(r) - 0.15$, where $r$ is the rating value. The results (a single trial) are reported in Table 8. We see our AEMC-NE outperforms AEMC in all cases.

Table 8: MCAR vs MNAR on MovieLens datasets

| data | missing mechanism | AEMC | AEMC-NE |
|------|------|------|---------|
| ML-100K | MCAR | 0.886 | 0.878 |
| ML-100K | MNAR | 0.988 | 0.920 |
| ML-1M | MCAR | 0.828 | 0.825 |
| ML-1M | MNAR | 0.902 | 0.853 |
| ML-10M | MCAR | 0.780 | 0.773 |
| ML-10M | MNAR | 0.830 | 0.815 |

## G    EXPERIMENTS OF MISSING DATA IMPUTATION ON UCI DATASETS

Here we consider general missing data imputation on four UCI[2] datasets, "Breast"[3], "Letter"[4], "Credit"[5], and "News". The results are reported in Table 9, where the missing rate is 20% and the results of MissForest, MICE, EM, Auto-encoder, and GAIN are all from (Yoon et al., 2018).

We see that our AEMC-NE outperforms all other methods on the Credit data and outperforms Auto-encoder in all cases. Note that in this study, we just set the numbers of hidden units in the two neural networks of AEMC-NE as 100 and 50 respectively and the data preprocessing is the same as that of (Yoon et al., 2018). It is expected that the performance of AEMC-NE can be further improved by tuning the numbers of hidden layers and hidden units.

Table 9: Experiment result on UCI datasets

| RMSE | Breast | Letter | Credit | News |
|------|--------|--------|--------|------|
| MissForest | 0.0608 | 0.1605 | 0.1976 | 0.1623 |
| MICE | 0.0646 | 0.1537 | 0.2585 | 0.1763 |
| EM | 0.0634 | 0.1563 | 0.2604 | 0.1912 |
| Auto-encoder | 0.0697 | 0.1351 | 0.2388 | 0.1667 |
| GAIN (Yoon et al., 2018) | 0.0546 | 0.1198 | 0.1858 | 0.1441 |
| **AEMC-NE** | 0.0634 | 0.1328 | 0.1775 | 0.1551 |

## H    PROOF FOR THEOREM 3.1

First of all, we give the following lemmas.

**Lemma H.1.** *Let $\mathcal{H} = \left\{ \boldsymbol{H} \in \mathbb{R}^{m \times n} : h_{ij} = \boldsymbol{\Theta}_{L_\Theta} \sigma \left( \boldsymbol{\Theta}_{L_\Theta - 1}(\cdots \sigma(\boldsymbol{\Theta}_1 z_{ij}) \cdots) \right), \forall (i,j) \in [m] \times [n]; \boldsymbol{\Theta}_l \in \mathbb{R}^{p_l \times p_{l-1}}, \|\boldsymbol{\Theta}_l\|_2 \leq b_l, \|\boldsymbol{\Theta}_l\|_F \leq b'_l, \forall l \in [L_\Theta]; \boldsymbol{Z} \in \mathcal{Z}, \|\boldsymbol{Z}\|_F \leq s_z \right\}$, where the Lipschitz constant of $\sigma$ is $\varrho$. Suppose the covering number of $\mathcal{Z}$ with respect to $\|\cdot\|_F$ is upper-bounded by $\kappa_\varepsilon$ and $\varepsilon = \epsilon \left( 2\varrho^{L_\Theta - 1} \prod_{l=1}^{L_\Theta} b_l \right)^{-1}$. Then the cover covering number of $\mathcal{H}$ with respect to $\|\cdot\|_F$ is bounded as*

$$\mathcal{N}(\mathcal{H}, \|\cdot\|_F, \epsilon) \leq \kappa_\varepsilon \prod_{l=1}^{L_\Theta} \left( \frac{3\sqrt{2}\varrho^{L_\Theta - 1}(L_\Theta + 1)s_z \prod_{l=1}^{L_\Theta} b_l}{\epsilon} \right)^{p_l p_{l-1}}.$$

---

[2]https://archive.ics.uci.edu/ml/index.php

[3]https://archive.ics.uci.edu/ml/datasets/breast+cancer+wisconsin+(diagnostic)

[4]https://archive.ics.uci.edu/ml/datasets/letter+recognition

[5]https://archive.ics.uci.edu/ml/datasets/credit+approval

*Proof.* See Section K.1. □

Lemma H.1 provides an upper bound of the covering number of the element-wise neural network. The following lemma shows an upper bound of the covering number of the main neural network.

**Lemma H.2** (Theorem 3.3 of (Bartlett et al., 2017), reformulated). *Let* $\mathbf{Z} = \mathbf{W}_{L_W} \sigma \left( \mathbf{W}_{L_W-1} \left( \cdots \sigma(\mathbf{W}_1 \tilde{\mathbf{X}}) \cdots \right) \right)$, *where* $\mathbf{W}_l \in \mathbb{R}^{d_{l+1} d_l}$, $l \in [L_W]$, *and* $\max(m, d_1, \ldots, d_{L_W}) \leq D$. *Denote the Lipschitz constant of* $\sigma$ *by* $\rho$. *Suppose the reference matrices* $(\mathbf{M}_1, \ldots, \mathbf{M}_{L_W})$ *are given. Define*

$$\mathcal{C} = \{F_{\mathcal{W}}(\tilde{\mathbf{X}}) : \mathcal{W} = (\mathbf{W}_1, \ldots, \mathbf{W}_{L_W}), \|\mathbf{W}_l\|_\sigma \leq a_l, \|\mathbf{W}_l - \mathbf{M}_l\|_\sigma \leq a'_l, \forall \in [L_W]\}.$$

*The for any* $\epsilon > 0$,

$$\ln \mathcal{N}\left(\mathcal{C}, \epsilon, \|\cdot\|_F\right) \leq \frac{\|\tilde{\mathbf{X}}\|_F^2 \ln 2D^2}{\epsilon^2} \left( \rho^{2(L_W-1)} \prod_{l=1}^{L_W} a_l^2 \right) \left( \sum_{l=1}^{L_W} \left( \frac{a'_l}{a_l} \right)^{2/3} \right)^3.$$

Now we can get an upper bound for the covering number of the entire neural network in NE-AEMC.

**Lemma H.3.** *The covering number of* $\mathcal{H}_{W,\Theta} = \{H_\Theta(F_{\mathcal{W}(\tilde{\mathbf{X}})})\}$ *with respect to* $\|\cdot\|_F$ *satisfies*

$$\ln \mathcal{N}(\mathcal{H}, \|\cdot\|_F, \epsilon) \leq \frac{4\varrho^{2(L_\Theta-1)} \rho^{2(L_W-1)} \|\tilde{\mathbf{X}}\|_F^2 \ln 2D^2}{\epsilon^2} \left( \prod_{l=1}^{L_W} a_l^2 \right) \left( \prod_{l=1}^{L_\Theta} b_l^2 \right) \left( \sum_{l=1}^{L_W} \left( \frac{a'_l}{a_l} \right)^{2/3} \right)^3$$
$$+ \left( \sum_{l=1}^{L_\Theta} p_l p_{l-1} \right) \ln \left( \frac{6 L_\Theta \varrho^{L_\Theta-1} \rho^{L_W-1} \|\tilde{\mathbf{X}}\|_F \left( \prod_{l=1}^{L_W} a_l \right) \left( \prod_{l=1}^{L_\Theta} b_l \right) \max_l \frac{b'_l}{b_l}}{\epsilon} \right).$$

*Proof.* See Section K.2. □

Now we can calculate the upper bound of the Rademacher complexity of $\mathcal{H}_{W,\Theta}$ via using Lemma H.3 and Dudley entropy integral bound.

**Lemma H.4.** *Let* $v_1 = 4\rho^{2(L_W-1)} \varrho^{2(L_\Theta-1)} \|\tilde{\mathbf{X}}\|_F^2 \ln 2D^2 \left( \prod_{l=1}^{L_W} a_l^2 \right) \left( \sum_{l=1}^{L_W} \left( \frac{a'_l}{a_l} \right)^{2/3} \right)^3 \left( \prod_{l=1}^{L_\Theta} b_l^2 \right)$,

$v_2 = \sum_{l=1}^{L_\Theta} p_l p_{l-1}$, *and* $v_3 = 6 L_\Theta \rho^{L_W-1} \varrho^{L_\Theta-1} \|\tilde{\mathbf{X}}\|_F \left( \prod_{l=1}^{L_W} a_l \right) \left( \prod_{l=1}^{L_\Theta} b_l \right) \max_l b'_l b_l^{-1}$. *Suppose* $\|h_\Theta(g_W(\tilde{\mathbf{X}}))\|_\infty \leq \mu$. *The Rademacher complexity of* $\mathcal{H}_{W,\Theta}$ *is bounded as*

$$\mathcal{R}_S(\mathcal{H}_{W,\Theta}) \leq \frac{4\mu}{S} + \frac{12\sqrt{v_1 + \mu^2 v_2} \ln S}{S} + \frac{12\mu\sqrt{v_2 \ln \mu^{-1} v_3}}{\sqrt{S}}. \tag{13}$$

*Proof.* See Section K.3. □

Collaborative filtering is a transductive learning problem. The following lemma provides a sample complexity bound for transductive learning, which is consistent with the objective function and evaluation metric (RMSE) widely used in collaborative filtering.

**Lemma H.5** (Theorem 1 of (El-Yaniv & Pechony, 2009), reformulated). *Let* $\mathcal{H}$ *be a fixed hypothesis set and suppose* $\sup_{i,j|\mathbf{X} \in \mathcal{H}} |\ell(Y_{ij}, X_{ij})| \leq \tau_\ell$. *Suppose a fixed set* $S$ *of distinct indices is uniformly and randomly split to two subsets* $S_{\text{train}}$ *and* $S_{\text{test}}$. *Then with probability at least* $1 - \delta$ *over the random split, we have*

$$\frac{1}{|S_{\text{test}}|} \sum_{(i,j) \in S_{\text{test}}} \ell(Y_{ij}, X_{ij}) \leq \frac{1}{|S_{\text{train}}|} \sum_{(i,j) \in S_{\text{train}}} \ell(Y_{ij}, X_{ij}) + \mathcal{R}_{|S_{\text{train}}|+|S_{\text{test}}|}(\ell \circ \mathcal{H})$$
$$+ \frac{11\tau_\ell (|S_{\text{train}}| + |S_{\text{test}}|) \sqrt{\min(|S_{\text{train}}|, |S_{\text{test}}|)}}{|S_{\text{train}}||S_{\text{test}}|}$$
$$+ 3\tau_\ell \sqrt{\frac{(|S_{\text{train}}| + |S_{\text{test}}|)}{|S_{\text{train}}||S_{\text{test}}|} \ln \frac{1}{\delta}} \tag{14}$$

It should be pointed out that according to (El-Yaniv & Pechyony, 2009), the transductive Rademacher complexity $\mathcal{R}_{|S_{\text{train}}|+|S_{\text{test}}|}$ has the following relationship with the inductive complexity $\mathcal{R}_S$ (to be computed later)

$$\mathcal{R}_{|S_{\text{train}}|+|S_{\text{test}}|}(\ell \circ \mathcal{H}) = \frac{(|S_{\text{train}}| + |S_{\text{test}}|)^2}{|S_{\text{train}}||S_{\text{test}}|}\mathcal{R}_S(\ell \circ \mathcal{H}). \tag{15}$$

Then Theorem 3.1 can be proved as follows.

*Proof.* Accoding to the Rademacher contraction property, we have $\mathcal{R}_S(\ell \circ \mathcal{H}) \leq \eta_\ell \mathcal{R}_S(\mathcal{H})$, where $\eta_\ell$ denotes the lipschitz constant of $\ell$. Using Lemma H.5 with a slightly different notation and Lemma H.4 (let $S = mn$) where $\mu^2 v_2 \ll v_1$ provided that the element-wise neural network is small enough, we have

$$
\begin{aligned}
\frac{1}{|S^c|}\sum_{(i,j)\in S^c} \ell\left(X_{ij}, \hat{X}_{ij}\right) \leq & \frac{1}{|S|}\sum_{(i,j)\in S} \ell\left(X_{ij}, \hat{X}_{ij}\right) \\
& + \eta_\ell \vartheta \left(\frac{4\mu}{mn} + \frac{12\sqrt{v_1 + \mu^2 v_2}\ln(mn)}{mn} + \frac{12\mu\sqrt{v_2 \ln \mu^{-1} v_3}}{\sqrt{mn}}\right) \\
& + \frac{11\tau_\ell\left(|S| + |S^c|\right)\sqrt{\min\left(|S|, |S^c|\right)}}{|S||S^c|} + 3\tau_\ell\sqrt{\frac{\left(|S| + |S^c|\right)}{|S||S^c|}\ln\frac{1}{\delta}} \\
\leq & \frac{1}{|S|}\sum_{(i,j)\in S} \ell\left(X_{ij}, \hat{X}_{ij}\right) \\
& + \frac{C_1\eta_\ell v_1' mn \ln(mn)}{|S||S^c|} + \frac{C_2\eta_\ell\mu(mn)^{3/2}}{|S||S^c|}\sqrt{v_2 \ln v_3'} \\
& + \frac{11\tau_\ell mn\sqrt{\min\left(|S|, |S^c|\right)}}{|S||S^c|} + 3\tau_\ell\sqrt{\frac{mn}{|S||S^c|}\ln\frac{1}{\delta}},
\end{aligned}
\tag{16}
$$

where $C_1$ and $C_2$ are some fixed constants,

$$v_1' = \rho^{(L_W - 1)}\varrho^{(L_\Theta - 1)}\|\boldsymbol{X}\|_F\sqrt{\ln D}\left(\prod_{l=1}^{L_W} a_l\right)\left(\sum_{l=1}^{L_W}\left(\frac{a_l'}{a_l}\right)^{2/3}\right)^{3/2}\left(\prod_{l=1}^{L_\Theta} b_l\right),$$

$v_3' = L_\Theta\mu^{-1}\gamma\rho^{L_W - 1}\varrho^{L_\Theta - 1}\|\boldsymbol{X}\|_F\left(\prod_{l=1}^{L_W} a_l\right)\left(\prod_{l=1}^{L_\Theta} b_l\right)\max_l \frac{b_l'}{b_l}$, and $\vartheta = \frac{(|S| + |S^c|)^2}{|S||S^c|} = \frac{m^2 n^2}{|S||S^c|}$. Note that $v_3' > 1$ holds trivially because $\|\tilde{\boldsymbol{X}}\|_F\left(\prod_{l=1}^{L_W} a_l\right)\left(\prod_{l=1}^{L_\Theta} b_l\right)/\mu > 1$. Rename $v_1'$ and $v_3'$ as $v_1$ and $v_3$ respectively, we finish the proof. $\qquad\square$

## I   PROOF FOR THEOREM 3.2

*Proof.* For convenience, let $\mathcal{L}(\hat{\boldsymbol{X}}) = \frac{1}{mn}\sum_{(i,j)\in[m]\times[n]}\ell(X_{ij}, \hat{X}_{ij})$, $\mathcal{L}_S^P(\hat{\boldsymbol{X}}) = \frac{1}{mn}\sum_{(i,j)\in S} p_{ij}^{-1}\ell(X_{ij}, \hat{X}_{ij})$, $\mathcal{L}(\hat{\boldsymbol{X}}') = \frac{1}{mn}\sum_{(i,j)\in[m]\times[n]}\ell(X_{ij}, \hat{X}_{ij}')$, and $\mathcal{L}_S^P(\hat{\boldsymbol{X}}') = \frac{1}{|S|}\sum_{(i,j)\in S}\frac{|S|}{mnp_{ij}}\ell(X_{ij}, \hat{X}_{ij}') = \frac{1}{mn}\sum_{(i,j)\in S} p_{ij}^{-1}\ell(X_{ij}, \hat{X}_{ij}')$.

$$|\mathcal{L}(\hat{\boldsymbol{X}}) - \mathcal{L}(\hat{\boldsymbol{X}}')|$$

$$= \left| \frac{1}{mn} \sum_{(i,j)\in[m]\times[n]} \left( \ell(X_{ij}, \hat{X}_{ij}) - \ell(X_{ij}, \hat{X}'_{ij}) \right) \right|$$

$$\leq \frac{1}{mn} \sum_{(i,j)\in[m]\times[n]} \left| \ell(X_{ij}, \hat{X}_{ij}) - \ell(X_{ij}, \hat{X}'_{ij}) \right| \tag{17}$$

$$\leq \frac{\eta_\ell}{mn} \sum_{(i,j)\in[m]\times[n]} \left| \hat{X}_{ij} - \hat{X}'_{ij} \right|$$

$$\leq \frac{\eta_\ell}{\sqrt{mn}} \|\hat{\boldsymbol{X}} - \hat{\boldsymbol{X}}'\|_F$$

$$|\mathcal{L}_S^P(\hat{\boldsymbol{X}}') - \mathcal{L}_S^P(\hat{\boldsymbol{X}})|$$

$$= \left| \frac{1}{mn} \sum_{(i,j)\in S} p_{ij}^{-1} \left( \ell(X_{ij}, \hat{X}'_{ij}) - \ell(X_{ij}, \hat{X}_{ij}) \right) \right|$$

$$\leq \frac{1}{mn} \sum_{(i,j)\in S} p_{ij}^{-1} \left| \ell(X_{ij}, \hat{X}'_{ij}) - \ell(X_{ij}, \hat{X}_{ij}) \right|$$

$$\leq \frac{\eta_\ell}{mn} \sum_{(i,j)\in S} p_{ij}^{-1} \left| \hat{X}'_{ij} - \hat{X}_{ij} \right| \tag{18}$$

$$\leq \frac{\eta_\ell}{mn} \sqrt{ \left( \sum_{(i,j)\in S} p_{ij}^{-2} \right) \left( \sum_{(i,j)\in S} \left| \hat{X}'_{ij} - \hat{X}_{ij} \right|^2 \right) }$$

$$\leq \frac{\eta_\ell \sqrt{\sum_{(i,j)\in S} p_{ij}^{-2}}}{mn} \|\hat{\boldsymbol{X}}' - \hat{\boldsymbol{X}}\|_F$$

According to the definitions of $\mathcal{L}_S^P(\hat{\boldsymbol{X}}')$ and $\mathcal{L}(\hat{\boldsymbol{X}}')$, we have $\mathbb{E}_S\left[\mathcal{L}_S^P(\hat{\boldsymbol{X}}')\right] = \mathcal{L}(\hat{\boldsymbol{X}}')$. We also have $\left| \frac{|S|}{mn p_{ij}} \ell(X_{ij}, \hat{X}_{ij}) \right| \leq \frac{\tau_\ell |S|}{mn p_{ij}} \triangleq \tau_{ij}$.

**Lemma I.1** ((Hoeffding, 1963))**.** *Let $\mathcal{X} = (x_1, x_2, \ldots, x_N)$ be a finite population of $N$ points and $X_1, X_2, \ldots, X_n$ be a random sample drawn without replacement from $\mathcal{X}$, where $a_i \leq X_i \leq b_i$, $i = 1, 2, \ldots, n$. Then for all $\varepsilon \geq 0$,*

$$\mathbb{P}\left[ \frac{1}{n} \sum_{i=1}^n X_i - \mu \geq \varepsilon \right] \leq \exp\left( -\frac{2n^2\varepsilon^2}{\sum_{i=1}^n (b_i - a_i)^2} \right) \tag{19}$$

*where $\mu = \frac{1}{N} \sum_{i=1}^N x_i$ is the mean of $\mathcal{X}$.*

According to Lemma I.1, we have

$$\mathbb{P}\left[ \left| \mathcal{L}(\hat{\boldsymbol{X}}') - \mathcal{L}_S^P(\hat{\boldsymbol{X}}') \right| \geq \varepsilon \right] \leq 2\exp\left( -\frac{|S|^2\varepsilon^2}{2\sum_{(i,j)\in S} \tau_{ij}^2} \right). \tag{20}$$

Using union bound for all $\hat{\boldsymbol{X}}' \in \mathcal{S}'$, we get

$$\mathbb{P}\left[ \sup_{\hat{\boldsymbol{X}}'\in\mathcal{S}'} \left| \mathcal{L}(\hat{\boldsymbol{X}}') - \mathcal{L}_S^P(\hat{\boldsymbol{X}}') \right| \geq \varepsilon \right] \leq 2|\mathcal{S}'|\exp\left( -\frac{|S|^2\varepsilon^2}{2\sum_{(i,j)\in S} \tau_{ij}^2} \right). \tag{21}$$

Letting $\varepsilon = \sqrt{\frac{2\sum_{(i,j)\in S}\tau_{ij}^2}{|S|^2}\ln(mn|\mathcal{S}'|)}$, then with probability at least $1 - \frac{2}{mn}$, we have

$$\sup_{\hat{\boldsymbol{X}}'\in\mathcal{S}'} \left| \mathcal{L}(\hat{\boldsymbol{X}}') - \mathcal{L}_S^P(\hat{\boldsymbol{X}}') \right| \leq \sqrt{\frac{2\sum_{(i,j)\in S}\tau_{ij}^2}{|S|^2}\ln(mn|\mathcal{S}'|)}. \tag{22}$$

Now with probability at least $1 - \frac{2}{mn}$, we have

$$
\sup_{\hat{\boldsymbol{X}} \in \mathcal{S}} \left| \mathcal{L}(\hat{\boldsymbol{X}}) - \mathcal{L}_S^P(\hat{\boldsymbol{X}}) \right|
$$

$$
\leq \sup_{\hat{\boldsymbol{X}} \in \mathcal{S}} \left| \mathcal{L}(\hat{\boldsymbol{X}}) - \mathcal{L}(\hat{\boldsymbol{X}}') \right| + \left| \mathcal{L}(\hat{\boldsymbol{X}}') - \mathcal{L}_S^P(\hat{\boldsymbol{X}}') \right| + \left| \mathcal{L}_S^P(\hat{\boldsymbol{X}}') - \mathcal{L}_S^P(\hat{\boldsymbol{X}}) \right|
$$

$$
\leq \sup_{\hat{\boldsymbol{X}} \in \mathcal{S}} \frac{\eta_\ell}{\sqrt{mn}} \|\hat{\boldsymbol{X}} - \hat{\boldsymbol{X}}'\|_F + \sqrt{\frac{2 \sum_{(i,j)\in S} \tau_{ij}^2}{|S|^2} \ln(mn|\mathcal{S}'|)} + \sup_{\hat{\boldsymbol{X}} \in \mathcal{S}} \frac{\eta_\ell \sqrt{\sum_{(i,j)\in S} p_{ij}^{-2}}}{mn} \|\hat{\boldsymbol{X}}' - \hat{\boldsymbol{X}}\|_F
$$

$$
\leq \eta_\ell \epsilon \left( \frac{1}{\sqrt{mn}} + \frac{\sqrt{\sum_{(i,j)\in S} p_{ij}^{-2}}}{mn} \right) + \frac{\tau_\ell \sqrt{\sum_{(i,j)\in S} p_{ij}^{-2}}}{mn} \sqrt{2 \ln(mn|\mathcal{S}'|)}.
$$

$$(23)$$

Note that $\ln|\mathcal{S}'| = \ln \mathcal{N}(\mathcal{H}, \|\cdot\|_F, \epsilon)$, then $\ln|\mathcal{S}'| \leq \frac{v_1}{\epsilon^2} + v_2 \ln \frac{v_3}{\epsilon}$, where

$v_1 = 4\rho^{2(L_W-1)} \varrho^{2(L_\Theta-1)} \|\tilde{\boldsymbol{X}}\|_F^2 \ln 2D^2 \left( \prod_{l=1}^{L_W} a_l^2 \right) \left( \sum_{l=1}^{L_W} \left( \frac{a_l'}{a_l} \right)^{2/3} \right)^3 \left( \prod_{l=1}^{L_\Theta} b_l^2 \right)$, $v_2 =$

$\sum_{l=1}^{L_\Theta} p_l p_{l-1}$, and $v_3 = 6 L_\Theta \rho^{L_W-1} \varrho^{L_\Theta-1} \|\tilde{\boldsymbol{X}}\|_F \left( \prod_{l=1}^{L_W} a_l \right) \left( \prod_{l=1}^{L_\Theta} b_l \right) \max_l b_l' b_l^{-1}$. Then we arrive at

$$
\sup_{\hat{\boldsymbol{X}} \in \mathcal{S}} |\mathcal{L}(\hat{\boldsymbol{X}}) - \mathcal{L}_S^P(\hat{\boldsymbol{X}})|
$$

$$
\leq \eta_\ell \epsilon \left( \frac{1}{\sqrt{mn}} + \frac{\sqrt{\sum_{(i,j)\in S} p_{ij}^{-2}}}{mn} \right) + \frac{\tau_\ell \sqrt{\sum_{(i,j)\in S} p_{ij}^{-2}}}{mn} \sqrt{2 \ln(mn) + \frac{2v_1}{\epsilon^2} + 2v_2 \ln \frac{v_3}{\epsilon}}
$$

$$
\leq \eta_\ell \epsilon \left( \frac{1}{\sqrt{mn}} + \frac{\sqrt{\sum_{(i,j)\in S} p_{ij}^{-2}}}{mn} \right) + \frac{\tau_\ell \sqrt{\sum_{(i,j)\in S} p_{ij}^{-2}}}{mn} \left( \sqrt{2 \ln(mn)} + \frac{\sqrt{2v_1}}{\epsilon} + \sqrt{2v_2 \ln \frac{v_3}{\epsilon}} \right)
$$

$$
\leq \eta_\ell \epsilon \left( \frac{1}{\sqrt{mn}} + \frac{\sqrt{\sum_{(i,j)\in S} p_{ij}^{-2}}}{mn} \right) + \frac{\tau_\ell \sqrt{\sum_{(i,j)\in S} p_{ij}^{-2}}}{mn} \left( \frac{C_1' \sqrt{v_1}}{\epsilon} + C_2' \sqrt{v_2 \ln \frac{v_3}{\epsilon}} \right)
$$

$$
\leq \tau_\ell \left( \frac{1}{\sqrt{mn}} + \frac{\sqrt{\sum_{(i,j)\in S} p_{ij}^{-2}}}{mn} \right) + \frac{C_1' \eta_\ell \sqrt{\sum_{(i,j)\in S} p_{ij}^{-2}} \sqrt{v_1}}{mn} + \frac{C_2' \tau_\ell \sqrt{\sum_{(i,j)\in S} p_{ij}^{-2}}}{mn} \sqrt{v_2 \ln \frac{\eta_\ell v_3}{\tau_\ell}}
$$

$$
\leq \tau_\ell \left( \frac{1}{\sqrt{mn}} + \frac{\sqrt{\sum_{(i,j)\in S} p_{ij}^{-2}}}{mn} \right) + \frac{C_1'' \eta_\ell \sqrt{\sum_{(i,j)\in S} p_{ij}^{-2}} v_1'}{mn} + \frac{C_2'' \tau_\ell \sqrt{\sum_{(i,j)\in S} p_{ij}^{-2}}}{mn} \sqrt{v_2 \ln \frac{\eta_\ell v_3'}{\tau_\ell}},
$$

$$(24)$$

where we have let $\epsilon = \frac{\tau_\ell}{\eta_\ell}$, $C_1'$, $C_1''$, $C_2'$, and $C_2''$ are some numerical constants, $v_1' = \sqrt{v_1}$, $v_3' = \mu^{-1} v_3$. Then we rename $v_1'$, $v_3'$, $C_1''$, and $C_2''$ as $v_1$, $v_3$, $C_1'$, and $C_2'$ respectively. This finished the proof.

$\square$

## J  PROOF FOR THEOREM A.1

*Proof.* Note that

$$
|\mathcal{L}(\hat{\boldsymbol{X}}) - \mathcal{L}_S^{\hat{P}}(\hat{\boldsymbol{X}})| \leq |\mathcal{L}(\hat{\boldsymbol{X}}) - \mathbb{E}_S[\mathcal{L}_S^{\hat{P}}(\hat{\boldsymbol{X}})]| + |\mathcal{L}_S^{\hat{P}}(\hat{\boldsymbol{X}}) - \mathbb{E}_S[\mathcal{L}_S^{\hat{P}}(\hat{\boldsymbol{X}})]|. \tag{25}
$$

Then we need to bound $|\mathcal{L}(\hat{\boldsymbol{X}}) - \mathbb{E}_S[\mathcal{L}_S^{\hat{P}}(\hat{\boldsymbol{X}})]|$ and $|\mathcal{L}_S^{\hat{P}}(\hat{\boldsymbol{X}}) - \mathbb{E}_S[\mathcal{L}_S^{\hat{P}}(\hat{\boldsymbol{X}})]|$ respectively.

We have

$$|\mathcal{L}(\hat{\boldsymbol{X}}) - \mathbb{E}_S[\mathcal{L}_S^{\hat{P}}(\hat{\boldsymbol{X}})]|$$

$$= \left| \frac{1}{mn} \sum_{(i,j)\in[m]\times[n]} \ell(X_{ij}, \hat{X}_{ij}) - \frac{1}{mn} \sum_{(i,j)\in[m]\times[n]} \frac{p_{ij}}{\hat{p}_{ij}} \ell(X_{ij}, \hat{X}_{ij}) \right|$$

$$= \frac{1}{mn} \left| \sum_{(i,j)\in[m]\times[n]} \left(1 - \frac{p_{ij}}{\hat{p}_{ij}}\right) \ell(X_{ij}, \hat{X}_{ij}) \right|$$

$$\leq \frac{\tau_\ell}{mn} \sum_{(i,j)\in[m]\times[n]} \left| 1 - \frac{p_{ij}}{\hat{p}_{ij}} \right|. \qquad (26)$$

It is worth noting that the procedures of bounding $|\mathcal{L}_S^{\hat{P}}(\hat{\boldsymbol{X}}) - \mathbb{E}_S[\mathcal{L}_S^{\hat{P}}(\hat{\boldsymbol{X}})]|$ are the same as that of $|\mathcal{L}(\hat{\boldsymbol{X}}) - \mathcal{L}_S^P(\hat{\boldsymbol{X}})|$ in the previous section. Therefore, we have

$$\left| \mathcal{L}_S^{\hat{P}}(\hat{\boldsymbol{X}}) - \mathbb{E}_S[\mathcal{L}_S^{\hat{P}}(\hat{\boldsymbol{X}})] \right|$$
$$\leq \frac{\sqrt{\sum_{(i,j)\in S} \hat{p}_{ij}^{-2}}}{mn} \left( C_1' \eta_\ell v_1 + C_2' \tau_\ell \sqrt{v_2 \ln v_3} \right) + \tau_\ell \left( \frac{1}{\sqrt{mn}} + \frac{\sqrt{\sum_{(i,j)\in S} \hat{p}_{ij}^{-2}}}{mn} \right), \qquad (27)$$

where meanings of $C_1'$, $C_2'$, $v_1$, $v_2$, and $v_3$ are the same as those in Theorem 3.2.

Now combining equation 26 and equation 27, we arrive at

$$|\mathcal{L}(\hat{\boldsymbol{X}}) - \mathcal{L}_S^{\hat{P}}(\hat{\boldsymbol{X}})|$$
$$\leq \frac{\sqrt{\sum_{(i,j)\in S} \hat{p}_{ij}^{-2}}}{mn} \left( C_1' \eta_\ell v_1 + C_2' \tau_\ell \sqrt{v_2 \ln v_3} \right) + \tau_\ell \left( \frac{1}{\sqrt{mn}} + \frac{\sqrt{\sum_{(i,j)\in S} \hat{p}_{ij}^{-2}}}{mn} \right)$$
$$+ \frac{\tau_\ell}{mn} \sum_{(i,j)\in[m]\times[n]} \left| 1 - \frac{p_{ij}}{\hat{p}_{ij}} \right|, \qquad (28)$$

which holds with probability at least $1 - \frac{2}{mn}$. This finished the proof. $\qquad\square$

## K  PROOF FOR LEMMAS

### K.1  PROOF FOR LEMMA H.1

*Proof.* Let $\mathcal{S}_{\Theta_l} := \{\boldsymbol{\Theta}_l \in \mathbb{R}^{p_{l+1}\times p_l} : \|\boldsymbol{\Theta}_l\|_2 \leq b_l, \|\boldsymbol{\Theta}_l\|_F \leq b_l'\}, \forall l \in [L_\Theta]$. It is known that there exists an $\epsilon_l$-net $\bar{\mathcal{S}}_{\Theta_l}$ obeying

$$\mathcal{N}(\mathcal{S}_{\Theta_l}, \|\cdot\|_F, \epsilon_l) \leq \left( \frac{3b_l'}{\epsilon_l} \right)^{p_l p_{l-1}}$$

such that $\|\boldsymbol{\Theta}_l - \bar{\boldsymbol{\Theta}}_l\|_F \leq \epsilon_l$. We have

$$
\begin{aligned}
|h_{ij} - \bar{h}_{ij}| &= \|h_{ij} - \bar{h}_{ij}\|_F \\
&= \left\| \boldsymbol{\Theta}_{L_\Theta} \sigma\left(\boldsymbol{\Theta}_{L_\Theta-1}(\cdots\sigma(\boldsymbol{\Theta}_1 z_{ij})\cdots)\right) - \bar{\boldsymbol{\Theta}}_{L_\Theta}\sigma\left(\bar{\boldsymbol{\Theta}}_{L_\Theta-1}(\cdots\sigma(\bar{\boldsymbol{\Theta}}_1 \bar{z}_{ij})\cdots)\right) \right\|_F \\
&= \Big\| \boldsymbol{\Theta}_{L_\Theta} \sigma\left(\boldsymbol{\Theta}_{L_\Theta-1}(\cdots\sigma(\boldsymbol{\Theta}_1 z_{ij})\cdots)\right) - \bar{\boldsymbol{\Theta}}_{L_\Theta}\sigma\left(\boldsymbol{\Theta}_{L_\Theta-1}(\cdots\sigma(\boldsymbol{\Theta}_1 z_{ij})\cdots)\right) \\
&\quad + \bar{\boldsymbol{\Theta}}_{L_\Theta}\sigma\left(\boldsymbol{\Theta}_{L_\Theta-1}(\cdots\sigma(\boldsymbol{\Theta}_1 z_{ij})\cdots)\right) - \bar{\boldsymbol{\Theta}}_{L_\Theta}\sigma\left(\bar{\boldsymbol{\Theta}}_{L_\Theta-1}(\cdots\sigma(\boldsymbol{\Theta}_1 z_{ij})\cdots)\right) + \cdots \\
&\quad + \bar{\boldsymbol{\Theta}}_{L_\Theta}\sigma\left(\bar{\boldsymbol{\Theta}}_{L_\Theta-1}(\cdots\sigma(\boldsymbol{\Theta}_1 z_{ij})\cdots)\right) - \bar{\boldsymbol{\Theta}}_{L_\Theta}\sigma\left(\bar{\boldsymbol{\Theta}}_{L_\Theta-1}(\cdots\sigma(\bar{\boldsymbol{\Theta}}_1 z_{ij})\cdots)\right) \Big\|_F \\
&\quad + \bar{\boldsymbol{\Theta}}_{L_\Theta}\sigma\left(\bar{\boldsymbol{\Theta}}_{L_\Theta-1}(\cdots\sigma(\bar{\boldsymbol{\Theta}}_1 z_{ij})\cdots)\right) - \bar{\boldsymbol{\Theta}}_{L_\Theta}\sigma\left(\bar{\boldsymbol{\Theta}}_{L_\Theta-1}(\cdots\sigma(\bar{\boldsymbol{\Theta}}_1 \bar{z}_{ij})\cdots)\right) \Big\|_F \\
&\leq \left\| \boldsymbol{\Theta}_{L_\Theta} \sigma\left(\boldsymbol{\Theta}_{L_\Theta-1}(\cdots\sigma(\boldsymbol{\Theta}_1 z_{ij})\cdots)\right) - \bar{\boldsymbol{\Theta}}_{L_\Theta}\sigma\left(\boldsymbol{\Theta}_{L_\Theta-1}(\cdots\sigma(\boldsymbol{\Theta}_1 z_{ij})\cdots)\right) \right\|_F \\
&\quad + \left\| \bar{\boldsymbol{\Theta}}_{L_\Theta}\sigma\left(\boldsymbol{\Theta}_{L_\Theta-1}(\cdots\sigma(\boldsymbol{\Theta}_1 z_{ij})\cdots)\right) - \bar{\boldsymbol{\Theta}}_{L_\Theta}\sigma\left(\bar{\boldsymbol{\Theta}}_{L_\Theta-1}(\cdots\sigma(\boldsymbol{\Theta}_1 z_{ij})\cdots)\right) \right\|_F + \cdots \\
&\quad + \left\| \bar{\boldsymbol{\Theta}}_{L_\Theta}\sigma\left(\bar{\boldsymbol{\Theta}}_{L_\Theta-1}(\cdots\sigma(\boldsymbol{\Theta}_1 z_{ij})\cdots)\right) - \bar{\boldsymbol{\Theta}}_L\sigma\left(\bar{\boldsymbol{\Theta}}_{L_\Theta-1}(\cdots\sigma(\bar{\boldsymbol{\Theta}}_1 z_{ij})\cdots)\right) \right\|_F \\
&\quad + \left\| \bar{\boldsymbol{\Theta}}_{L_\Theta}\sigma\left(\bar{\boldsymbol{\Theta}}_{L_\Theta-1}(\cdots\sigma(\bar{\boldsymbol{\Theta}}_1 z_{ij})\cdots)\right) - \bar{\boldsymbol{\Theta}}_{L_\Theta}\sigma\left(\bar{\boldsymbol{\Theta}}_{L_\Theta-1}(\cdots\sigma(\bar{\boldsymbol{\Theta}}_1 \bar{z}_{ij})\cdots)\right) \right\|_F \\
&\stackrel{(a)}{\leq} \varrho^{L_\Theta-1}|z_{ij}|\left\|\boldsymbol{\Theta}_{L_\Theta} - \bar{\boldsymbol{\Theta}}_{L_\Theta}\right\|_F \prod_{l=1}^{L_\Theta-1}\|\boldsymbol{\Theta}_l\|_2 \\
&\quad + \varrho^{L_\Theta-1}|z_{ij}|\left\|\bar{\boldsymbol{\Theta}}_{L_\Theta}\right\|_2\left\|\boldsymbol{\Theta}_{L_\Theta-1} - \bar{\boldsymbol{\Theta}}_{L_\Theta-1}\right\|_F \prod_{l=1}^{L_\Theta-2}\|\boldsymbol{\Theta}_l\|_2 + \cdots \\
&\quad + \varrho^{L-1}|z_{ij}|\left(\prod_{l=2}^{L_\Theta}\left\|\bar{\boldsymbol{\Theta}}_l\right\|_2\right)\left\|\boldsymbol{\Theta}_1 - \bar{\boldsymbol{\Theta}}_1\right\|_F \\
&\quad + \varrho^{L_\Theta-1}\|z_{ij} - \bar{z}_{ij}\|_F \prod_{l=1}^{L_\Theta}\|\bar{\boldsymbol{\Theta}}_l\|_2 \\
&\leq \varrho^{L_\Theta-1}\left( s_{z_{ij}}\epsilon_{L_\Theta}\prod_{l\neq L_\Theta} b_l + s_{z_{ij}}\epsilon_{L_\Theta-1}\prod_{l\neq L_\Theta-1} b_l + s_{z_{ij}}\epsilon_1 \prod_{l\neq 1}^{L_\Theta} b_l + \cdots + \|z_{ij} - \bar{z}_{ij}\|_F \prod_{l=1}^{L_\Theta} b_l \right).
\end{aligned}
$$
(29)

In (a), we used the facts $\|\boldsymbol{X}\boldsymbol{y}\|_F \leq \|\boldsymbol{X}\|_2\|\boldsymbol{y}\|_F$ and $\|\sigma(\boldsymbol{X}) - \sigma(\boldsymbol{Y})\|_F \leq \varrho\|\boldsymbol{X} - \boldsymbol{Y}\|_F$ recursively. It follows that

$$
\begin{aligned}
\|\boldsymbol{H} - \bar{\boldsymbol{H}}\|_F &= \sqrt{\sum_{ij}(h_{ij} - \bar{h}_{ij})^2} \\
&\stackrel{(a)}{\leq} \varrho^{L_\Theta-1}\sqrt{\sum_{ij} 2\left( s_{z_{ij}}^2\left(\epsilon_{L_\Theta}\prod_{l\neq L_\Theta} b_l + \epsilon_{L_\Theta-1}\prod_{l\neq L_\Theta-1} b_l + \cdots + \epsilon_1\prod_{l\neq 1}^{L_\Theta} b_l\right)^2 + \|z_{ij} - \hat{z}_{ij}\|_F^2\left(\prod_{l=1}^{L_\Theta} b_l\right)^2\right)} \\
&= \sqrt{2}\varrho^{L_\Theta-1}\sqrt{\left(\epsilon_{L_\Theta}\prod_{l\neq L_\Theta} b_l + \epsilon_{L_\Theta-1}\prod_{l\neq L_\Theta-1} b_l + \cdots + \epsilon_1\prod_{l\neq 1}^{L_\Theta} b_l\right)^2 \|\boldsymbol{Z}\|_F^2 + \left(\prod_{l=1}^{L_\Theta} b_l\right)^2 \|\boldsymbol{Z} - \bar{\boldsymbol{Z}}\|_F^2} \\
&\leq \sqrt{2}\varrho^{L_\Theta-1}\sqrt{\left(\epsilon_{L_\Theta}\prod_{l\neq L_\Theta} b_l + \epsilon_{L_\Theta-1}\prod_{l\neq L_\Theta-1} b_l + \cdots + \epsilon_1\prod_{l\neq 1}^{L_\Theta} b_l\right)^2 s_z^2 + \left(\prod_{l=1}^{L_\Theta} b_l\right)^2 \epsilon_z^2}.
\end{aligned}
$$
(30)

In (a), we used the fact $(x+y)^2 \leq 2(x^2+y^2)$. Let $\epsilon_l = \dfrac{\epsilon/(\sqrt{2}L_\Theta)}{\sqrt{2}\varrho^{L_\Theta-1}s_z\prod_{k\neq l} b_k}, \forall l \in [L_\Theta]$. Let $\epsilon_z = \dfrac{\epsilon/\sqrt{2}}{\sqrt{2}\varrho^{L_\Theta-1}\prod_{l=1}^{L_\Theta} b_l}$. We arrive at

$$
\|\boldsymbol{H} - \bar{\boldsymbol{H}}\|_F \leq \epsilon.
$$
(31)

It means that $\bar{\boldsymbol{H}}$ is an $\epsilon$-cover of $\boldsymbol{H}$. Then the covering number of $\mathcal{H}$ is bounded as

$$
\begin{aligned}
&\mathcal{N}(\mathcal{H}, \|\cdot\|_F, \epsilon) \\
&\leq \mathcal{N}(\mathcal{Z}, \|\cdot\|_F, \epsilon_z) \prod_{l=1}^{L_\Theta} \mathcal{N}(\mathcal{S}_{\Theta_l}, \|\cdot\|_F, \epsilon_l) \\
&\leq \kappa_\varepsilon \prod_{l=1}^{L_\Theta} \left( \frac{6\varrho^{L_\Theta-1} L_\Theta s_z b_l' \prod_{k\neq l} b_k}{\epsilon} \right)^{p_l p_{l-1}} \\
&= \kappa_\varepsilon \prod_{l=1}^{L_\Theta} \left( \frac{6\varrho^{L_\Theta-1} L_\Theta s_z b_l' b_l^{-1} \prod_{k=1}^{L_\Theta} b_k}{\epsilon} \right)^{p_l p_{l-1}} \\
&\leq \kappa_\varepsilon \left( \frac{C_\Theta}{\epsilon} \right)^{\sum_{l=1}^{L_\Theta} p_l p_{l-1}},
\end{aligned}
\tag{32}
$$

where $C_\Theta = 6\varrho^{L_\Theta-1} L_\Theta s_z \gamma \prod_{l=1}^{L_\Theta} b_l$ and $\gamma = \max_l b_l' b_l^{-1}$. This finished the proof. $\qquad\square$

## K.2 PROOF FOR LEMMA H.3

*Proof.* It is easy to show that $s_z$ can be determined by the following derivation

$$
\|\boldsymbol{Z}\|_F = \left\| \boldsymbol{W}_{L_W} \left( \sigma_W \left( \boldsymbol{W}_{L_W-1} \sigma_W (\cdots \sigma_W (\boldsymbol{W}_1 \tilde{\boldsymbol{X}}) \cdots) \right) \right) \right\|_F
\tag{33}
$$

$$
\leq \rho^{L_W-1} \|\boldsymbol{X}\|_F \prod_{l=1}^{L_W} a_l \triangleq s_z.
\tag{34}
$$

Combining Lemma H.1 and Lemma H.2, we have

$$
\begin{aligned}
&\ln \mathcal{N}(\mathcal{H}, \|\cdot\|_F, \epsilon) \\
&\leq \ln \kappa_\varepsilon + \left( \sum_{l=1}^{L_\Theta} p_l p_{l-1} \right) \ln \left( \frac{C_\Theta}{\epsilon} \right) \\
&\leq \frac{4\rho^{2(L_W-1)} \varrho^{2(L_\Theta-1)} \|\boldsymbol{X}\|_F^2 \ln 2D^2}{\epsilon^2} \left( \prod_{l=1}^{L_W} a_l^2 \right) \left( \sum_{l=1}^{L_W} \left( \frac{a_l'}{a_l} \right)^{2/3} \right)^3 \left( \prod_{l=1}^{L_\Theta} b_l^2 \right) \\
&\quad + \left( \sum_{l=1}^{L_\Theta} p_l p_{l-1} \right) \ln \left( \frac{6 L_\Theta \varrho^{L_\Theta-1} \rho^{L_W-1} \|\boldsymbol{X}\|_F \left( \prod_{l=1}^{L_W} a_l \right) \left( \prod_{i=1}^{L_\Theta} b_l \right) \max_l \frac{b_l'}{b_l}}{\epsilon} \right).
\end{aligned}
\tag{35}
$$

$\qquad\square$

## K.3 PROOF FOR LEMMA H.4

Before proof, we give the following lemma, which is a variant of the Dudley entropy integral bound on Rademacher complexity. Before proof, we give the following lemma, which is a variant of the Dudley entropy integral bound on Rademacher complexity.

**Lemma K.1** (Theorem 3 of (Schreuder, 2020)). *Let $\mathcal{F} \subset \{f : \mathcal{X} \mapsto \mathbb{R}\}$ be any class of measurable functions containing the uniformly zero function and let $B_S(\mathcal{F}) = \sup_{f \in \mathcal{F}} \|f\|_{L_2(P_S)}$. Then*

$$
\mathcal{R}_S(\mathcal{F}) \leq \inf_{\alpha > 0} \left( 4\alpha + \frac{12}{\sqrt{S}} \int_\alpha^{B_S(\mathcal{F})} \sqrt{\log \mathcal{N}(\mathcal{F}, L_2(P_S), \zeta)} \, \mathrm{d}\zeta \right).
\tag{36}
$$

In (Schreuder, 2020), $\|f\|_{L_2(P_S)}$ is defined as $\sqrt{\|f\|_{L_2(P_S)} = \frac{1}{S} \sum_{i=1}^{S} f(X_i)^2}$, which implies

$$
\mathcal{N}(\mathcal{F}, L_2(P_n), \zeta) = \mathcal{N}(\mathcal{F}, \|\cdot\|_F, \sqrt{n}\zeta).
\tag{37}
$$

It follows that

$$\mathcal{R}_S(\mathcal{F}) \leq \inf_{\alpha>0} \left( 4\alpha + \frac{12}{\sqrt{S}} \int_\alpha^{B_S(\mathcal{F})} \sqrt{\log\mathcal{N}(\mathcal{F}, \|\cdot\|_F, \sqrt{S}\zeta)} d\zeta \right)$$

$$= \inf_{\alpha>0} \left( \frac{4\alpha}{\sqrt{S}} + \frac{12}{S} \int_\alpha^{B_S(\mathcal{F})\sqrt{S}} \sqrt{\log\mathcal{N}(\mathcal{F}, \|\cdot\|_F, \epsilon)} d\epsilon \right). \tag{38}$$

Now we prove Lemma H.4.

*Proof.* For convenience, let

$$v_1 = 4\rho^{2(L_W-1)}\varrho^{2(L_\Theta-1)}\|\boldsymbol{X}\|_F^2 \ln 2D^2 \left(\prod_{l=1}^{L_W} a_l^2\right) \left(\sum_{l=1}^{L_W} \left(\frac{a_l'}{a_l}\right)^{2/3}\right)^3 \left(\prod_{l=1}^{L_\Theta} b_l^2\right),$$

$v_2 = \sum_{l=1}^{L_\Theta} p_l p_{l-1}$, and $v_3 = 6L_\Theta \rho^{L_W-1}\varrho^{L_\Theta-1}\|\boldsymbol{X}\|_F \left(\prod_{l=1}^{L_W} a_l\right)\left(\prod_{l=1}^{L_\Theta} b_l\right)\max_l b_l' b_l^{-1}$. Then $\ln\mathcal{N}(\mathcal{H}, \|\cdot\|_F, \epsilon) \leq \frac{v_1}{\epsilon^2} + v_2 \ln\left(\frac{v_3}{\epsilon}\right)$ from (35). Let $\mu = \max_{H\in\mathcal{H}} \|H\|_\infty$. According to equation 38, we have

$$\mathcal{R}_S(\mathcal{H}) \leq \inf_{\alpha>0} \left( \frac{4\alpha}{\sqrt{S}} + \frac{12}{S} \int_\alpha^{\mu\sqrt{S}} \sqrt{\frac{v_1}{\epsilon^2} + v_2 \ln\left(\frac{\mu^{-1}v_3}{\mu^{-1}\epsilon}\right)} d\epsilon \right)$$

$$\overset{(a)}{\leq} \inf_{\alpha>0} \left( \frac{4\alpha}{\sqrt{S}} + \frac{12}{S} \int_\alpha^{\mu\sqrt{S}} \sqrt{\frac{v_1 + \mu^2 v_2}{\epsilon^2} + v_2 \ln\mu^{-1}v_3} d\epsilon \right)$$

$$\overset{(b)}{\leq} \inf_{\alpha>0} \left( \frac{4\alpha}{\sqrt{S}} + \frac{12}{S} \int_\alpha^{\mu\sqrt{S}} \left( \frac{\sqrt{v_1 + \mu^2 v_2}}{\epsilon} + \sqrt{v_2 \ln\mu^{-1}v_3} \right) d\epsilon \right) \tag{39}$$

$$= \inf_{\alpha>0} \left( \frac{4\alpha}{\sqrt{S}} + \frac{12}{S} \left( \sqrt{v_1 + \mu^2 v_2} \ln\frac{\mu\sqrt{S}}{\alpha} + \sqrt{v_2 \ln\mu^{-1}v_3} \left(\mu\sqrt{S} - \alpha\right) \right) \right)$$

$$\overset{(c)}{\leq} \frac{4\mu}{S} + \frac{12\sqrt{v_1 + \mu^2 v_2}}{S} \ln S + \frac{12\mu(S-1)\sqrt{v_2 \ln\mu^{-1}v_3}}{S\sqrt{S}}$$

$$\leq \frac{4\mu}{S} + \frac{12\sqrt{v_1 + \mu^2 v_2} \ln S}{S} + \frac{12\mu\sqrt{v_2 \ln\mu^{-1}v_3}}{\sqrt{S}}.$$

In (a), we used the fact $\ln\frac{1}{x} \leq \frac{1}{x^2}$. The inequality (b) holds according to $\sqrt{x+y} \leq \sqrt{x} + \sqrt{y}$. In (c), we have let $\alpha = \frac{\mu}{\sqrt{S}}$, which though may not be the best choice. We arrive at

$$\mathcal{R}_S(\mathcal{H}_{W,\Theta}) \leq \frac{4\mu}{S} + \frac{12\sqrt{v_1 + \mu^2 v_2} \ln S}{S} + \frac{12\mu\sqrt{v_2 \ln\mu^{-1}v_3}}{\sqrt{S}}.$$

$\square$

