# OpenReview forum: "Neuron-Enhanced AutoEncoder Matrix Completion and Collaborative Filtering: Theory and Practice"
_ICLR.cc/2024/Conference — ICLR 2024 poster_

### Official Review · Reviewer_THxa · 2023-10-26

**Soundness:** 3 good
**Presentation:** 3 good
**Contribution:** 3 good
**Rating:** 8
**Confidence:** 4

**Summary:**

The paper proposed an autoencoder method for matrix completion and applied it to collaborative filtering. The corresponding generalization error bounds are also provided. The experiments showed the superiority of the proposed method over the baselines.

**Strengths:**

* This is a good work with new method, theory, and extensive experiments (many datasets, settings, baselines, and figures).
* The motivation is clear and the theoretical analysis, such as the bounds related to MCAR and MNAR, are insightful and practical for the recommendation systems.
* The proposed AEMC-NE is able to outperform SVD++, LLORMA, AutoRec, DMF+, etc.

**Weaknesses:**

* The time cost of the proposed method AEMC-NE is higher than that of AEMC.
* The motivation of conducting the experiments of Table 4 is not clear.
* Some important results (e.g., time costs comparison, Theorem A.1) are presented in the appendix, not the main paper. Theorem A.1 is more useful than Theorem 3.2 because the former is based on the estimated probability.

**Questions:**

1. According to equation (1), is the model $f$ actually a denoising autoencoder, with a partial reconstruction loss, not a classical autoencoder?
2. How does the data dimension $m$ affect the generalization error bounds? The authors only discussed $n$, the number of samples.
3. Regarding Theorem 3.2, if $p_{ij}$ is close to zero, is the bound useless? Could the authors give more explanation about the roles of $p_{ij}^{-2}$?
4. Are the values in Table 1 RMSEs?
5. In Figure 2, it seems that if the middle layer width of the main network is set to 10, the true latent dimension of the data, the performance of AEMC-NE is not better than those with larger width. What is the possible reason?
6. As shown by Table 2 and Table 3, the proposed AEMC-NE outperformed AEMC with ReLU output. Did the authors test AEMC with other activation function, such as sigmoid, in the output layer? How about their performance?
7. In Table 5, as the observation probabilities are known, what objective function have the authors used in the proposed method? The one in (9) or (7)?

---

> ### Author Response · Authors · 2023-11-16
> **Authors' response**
>
> Thanks for your comments. Our responses to Weaknesses and Questions are as follows.
>
> **Response to Weaknesses**:
> 1. Yes, our method is slightly slower than AEMC. However, the accuracy of our method is higher than that of AEMC in all cases of the experiments. In many real scenarios, a slightly higher time cost is acceptable as we require high accuracy.
> 2. In fact, we have explained the motivation in Section 5.4. In Tables 2 and 3, the rating matrices are close to square matrices and hence cannot fully demonstrate the superiority of our method over classical matrix factorization methods. Therefore, in Table 4, we use a subset of MovieLens-1M, where the matrix size is $3706\times 500$. In this matrices, $n\gg m$. According to our theorems, when $n$ is larger (compared to fixed network architecture), the generalization bound is tighter. Indeed, as shown by Table 4, the improvement over matrix factorization becomes more significant.
> 3. This is due to the page length limitation. We have tried our best to include more important results in the main paper.
>
>
> **Response to Questions**:
> 1. Yes, it is not a standard autoencoder. However, it is still a neural network to reconstruct the input (though partially). That's why we still call it autoencoder.
> 2. Thanks for raising the question. The reason we didn't discuss it is that when $m$ changes, the size of the neural network will change, which influences many terms in the error bound, e.g. the number of parameters, the missing rate, and $\Vert\tilde{\boldsymbol{X}}\Vert_F$. It is difficult to directly analyze the impact of $m$. If we fix the size of the hidden layers and the missing rate and assume that the latent dimension of the data does not change, then $m$ has little impact on the bound.
> 3. If $p_{ij}$ is close to zero, we may not observe the corresponding entry of the matrix. So in practice, $p_{ij}$ stays far from zero. Note that the bound is related to $\sqrt{\sum_{(i, j) \in S} p_{i j}^{-2}} /(m n)$, where there is a denominator $mn$. Thus, a few small $p_{ij}$ may not have a big impact on the bound.
> 4. No. The values in Table 1 are the relative recovery errors defined in the first paragraph of Section 5.1.
> 5. Thanks for the insightful comment. The possible reason is that when the latent dimension is too small, it will be difficult to train the two autoencoders of our method.  Figure 3(c) shows that AEMC is not sensitive to the latent dimension if it is larger than or equal to 10. The optimization of AEMC is easier than that of our AEMC-NE.
> 6. Sure, we tested Sigmoid and Tanh but their performances are much worse than that of ReLU.
> 7. We tried both (7) and (9) and found that their performances are similar. This also verified that our method AEMC-NE is useful for both MCAR (missing completely at random) and MNAR (missing not at random).

---

### Official Review · Reviewer_fr12 · 2023-10-27

**Soundness:** 4 excellent
**Presentation:** 3 good
**Contribution:** 3 good
**Rating:** 6
**Confidence:** 5

**Summary:**

The paper presented a neuron-enhanced autoencoder based method for matrix completion. The main idea is to adds an element-wise autoencoder to each output of the main autoencoder to enhance the reconstruction. The paper has both theoretical results (generalization bounds) and numerical results (collaborative filtering).

**Strengths:**

1. The proposed network architecture is composed of a main autoencoder and an element-wise. The idea is new.

2. There are two theorems of generalization analysis. One is for the case of missing completely with random, and the other is for the case of missing not at random. They verified the effectiveness of the proposed method theoretically.

3. The paper included a lot of experiments and the proposed method outperformed many baselines in most cases.

**Weaknesses:**

1. The writing style could be improved. Currently, the section of experimental evaluation focuses on collaborative filtering. To be consistent, the section of introduction should provide more information or discussion about collaborative filtering rather than matrix completion.

2. The time costs haven’t been compared, though the theoretical analysis was provided at the bottom of page 4.

**Questions:**

1. In Eq.(2), the loss function is the square loss. Has the authors tried other loss functions such as the absolute loss? Is the proposed method applicable to the setting of implicit feedback?

2. In Figure 1, is there an activation function in the output layer of the main network?

3. In Theorem 3.1, when $|S|$ or $|S^c|$ is close to zero, the bound is very loose. How to explain this phenomenon?

4. The results in Figure 2 look nice. But the motivation for the specific network architectures such as 300-100-30-100-300 should be illustrated.

---

> ### Author Response · Authors · 2023-11-16
> **Authors' response**
>
> Thank you very much for the positive assessment on our work.  Our responses are as follows.
>
> **Response to Weaknesses**: Thanks for pointing out the weaknesses. In the revised paper, we added more context about collaborative filtering to Introduction (e.g. paragraph 1).
> Actually, we have provided the time cost comparison in the original submission. It was shown by Table 6 of the appendix (Appendix E).
>
> **Response to Question 1**:
> Yes, we actually tested other loss functions. To ensure consistence with the evaluation metric RMSE, we use the square loss. If the evaluation metric is MAE, we’d better use absolute loss. By the way, the square loss is the most widely-used loss in collaborative filtering and matrix completion.
> Our method can be extended to the scenarios of implicit feedback if we use negative sampling and other loss functions such as BPR loss (Bayesian Personalized Ranking, S Rendle 2012). We added this discussion to the conclusion section of our paper.
>
> **Response to Question 2**:
> No. Our element-wise network can learn an activation function adaptively.
>
> **Response to Question 3**:
> This is inevitable in the setting of transductive learning. When $|S|$ or $|S^c|$ is close to zero, the uncertainty for the training error or test error will increase significantly.
>
> **Response to Question 4**:
> Thanks for pointing out it. This architecture was motivated by the fact that the dimension of the synthetic data is 300, the latent dimension is 10, and in practice we may never know the true latent dimension. Thus we use a larger latent dimension 30 in the autoencoder. Figure 2(c) also showed the effectiveness of the larger latent dimension.

---

### Official Review · Reviewer_fbH8 · 2023-10-28

**Soundness:** 4 excellent
**Presentation:** 4 excellent
**Contribution:** 3 good
**Rating:** 8
**Confidence:** 5

**Summary:**

The paper studies the problem of matrix completion with neural network, which is quite different from classical low-rank matrix completion. The authors proposed a new method as well as theoretical analysis. The experiments on synthetic data and real data demonstrated showed the usefulness of the proposed method.

**Strengths:**

The theory of neural network based collaborative filtering hasn’t been well-studied in literature. This paper proposed a new autoencoder matrix completion method and analyzed the generalization ability for different missing mechanisms. The strengths of the paper are as follows.
* The paper is well-organized. The literature review and related work are complete.
* The paper provided two theorems. The analysis and results are solid.
* The experiments are sufficient. The new method has been compared with more than five baselines on synthetic data, MovieLens 100k/1M/10, Doban and Flixter. And the performance is good.

**Weaknesses:**

Several minor issues are as follows.
* It is not very clear how to determine the architectures of the two autoencoders. Is there a rule of thumb?
* Some claims should be explained more detailedly. For instance, in the discussion for Theorem 3.2, more explanation about ‘$\sqrt{\sum_{(i, j) \in S} p_{i j}^{-2}} /(m n)$ is at the scale of $1 / \sqrt{|S|}$’ should be provided. In Table 2, a few results are missing. Why?

**Questions:**

* Do the two theorems provide any guidance for determine the architectures of the two autoencoders?
* According to the formulation of the proposed method, both the layer-wise network and the element-wise network can be any MLPs, not necessarily autoencoders. Right?
* In Analysis B of Section 3.1, is it possible that mean filling is better than zero filling for $\tilde{X}$?
* In Analysis C of Section 3.1, the authors only discussed the impact of $n$ on the generalization bound. What is the impact of $m$? A larger $m$ leads to a tighter bound?
* In Section 5.1, the latent dimension of the data generating model is 10, but as shown by the caption of Figure 2, the latent dimension of the layer-wise autoencoder was set to 30. What is the motivation of this setting?
* How to apply the proposed method to the collaborative filtering problem in the case of implicit feedback?

---

> ### Author Response · Authors · 2023-11-16
> **Authors' response**
>
> Thank you so much for the very positive evaluation. Our responses to your comments and questions are as follows.
>
> **Response to Weaknesses**:
> 1. Theoretically, according to our theorems, the size of the neural networks should not be too large when compared to the size of the training data. The number of the parameters of the two autoencoders should be smaller than the number of observed elements in the rating matrix. Empirically, we do not use deep autoencoders unless the number ($n$) of samples is much larger than the number ($m$) of variables (features). For instance, for the real datasets considered in our paper, the main autoencoder has only one hidden layer and the element-wise autoencoder also has only one hidden layer. For the synthetic data, because the number of samples (3000) is much larger than the number of features (300), as shown by the caption of Figure 2, we use three hidden layers in the main autoencoder and two hidden layers in the element-wise autoencoder. Figure 2(b) shows that our method is not sensitive to the width of the hidden layers of the element-wise autoencoder. In sum, when $n$ is much larger than $m$, we use more hidden layers in the two autoencoders.
> 2. The reason is that the average of $p_{ij}$ is about $\frac{|S|}{mn}$. We have improved the explanations in the revised paper. In Table 2, some values are missing because in the original papers of the corresponding methods, the datasets were not tested.
>
> **Response to Questions**:
> 1. Yes. This has been explained by our response to the Weakness 1.
> 2. Thanks for pointing out this property. Indeed, the two neural networks are not necessarily autoencoders. Their architectures can be asymmetric. Our theoretical results apply to general MLPs.
> 3. It is possible but it depends on data. In the theorem, when using zero filling, $\Vert\tilde{\boldsymbol{X}}\Vert_F$ can be reduced significantly; when using mean filling, $\Vert\tilde{\boldsymbol{X}}\Vert_F$ is large. If mean filling cannot reduce the training error significantly, the corresponding overall generalization bound will be looser.
> 4. The reason we didn't discuss $m$ is that when $m$ changes, the size of the neural network will change, which influences many terms in the error bound, e.g. the number of parameters, the missing rate, and $\Vert\tilde{\boldsymbol{X}}\Vert_F$. If we fix the size of the hidden layers and the missing rate and assume that the latent dimension of the data does not change, then $m$ has little impact on the bound.
> 5. There are two reasons for this setting. First, in real datasets, the true latent dimension of data is unknown. Thus we do not expect to use the true dimension in the experiment. Second, we find that, as shown by Figure 2(c), a larger $\hat{d}$ often leads to a better performance.
> 6. For implicit feedback, we need to use negative sampling and ranking loss functions such as BPR loss (Bayesian Personalized Ranking, S Rendle 2012). Our theoretical results apply to any loss functions. We added this discussion to the conclusion section of our paper.

---

### Official Review · Reviewer_uVGT · 2023-10-31

**Soundness:** 3 good
**Presentation:** 3 good
**Contribution:** 4 excellent
**Rating:** 8
**Confidence:** 5

**Summary:**

This work proposes a neuron-enhanced autoencoder matrix completion (AEMC-NE) method and applies it to collaborative filtering. AEMC-NE can adaptively learn an activation function for the output layer to approximate possibly complicated response functions in real data. The theoretical analysis and numerical results demonstrate the effectiveness of AEMC-NE in comparison to other methods.

**Strengths:**

1. The proposed algorithm is interesting and effective. It has lower RMSE than the compared method in the experiments of many benchmark datasets.
2. The theoretical results are rich and nontrivial. Theorem 3.1 and Theorem 3.2 shows the generalization error bounds of matrix completion in the settings of MCAR and MNAR respectively. An appealing theoretical result is that ‘Filling the missing values with zeros is helpful’. This is an important result for collaborative filtering.

**Weaknesses:**

Please refer to my questions.

**Questions:**

1. As shown in Fig.1, for the autoencoder, items correspond to samples while users correspond to features. Can we exchange the roles of items and users? What is the possible impact on the recommendation performance?
2. On page 4, it is stated that when the mechanism is missing not at random, the $Q$ in the objective function should be replaced by $S$. How to achieve $S$?
3. On page 7, is it possible that $\sqrt{\frac{L_W^3 \zeta \bar{d}}{n}}$ is larger than 1? It would be much better if the authors can provide an example of the value of $L_W^3 \zeta \bar{d}$.

---

> ### Author Response · Authors · 2023-11-16
> **Authors' response**
>
> Thanks a lot for recognizing our work. Our responses to your questions are as follows.
>
> **Response to Question 1**: Fig. 1 just shows an example. We can exchange the roles of items and users. As mentioned in Analysis C of Section 3.1, increasing $n$ (the number of samples) will reduce the generalization error bound and we hence propose to construct an autoencoder for the users when there are more users than items and vice versa.  In Table 4, we reported the experimental results of a few baselines on a subset of the MovieLens-1M dataset. The matrix size of the subset is 3706x500 (3706 items and 500 users), where the autoencoder of our method is for the items. We see the improvement over the matrix factorization method is more significant that that in Table 2 (entire MovieLens-1M). Actually, if the autoencoder is for users, the performance will degrade. This verifies the impact of the roles of items and users.
>
>
> **Response to Question 2**: As stated in Section 3.2, $Q$ can be obtained by a few approaches (Wang et al., 2019; Ma & Chen, 2019) such as Naive Bayes or Logistic Regression (Schnabel et al., 2016).
>
> **Response to Question 3**:  In $\sqrt{\frac{L_W^3 \xi_{\bar{d}}}{n}}$, $L_W$ is the number of layers of the network, $\bar{d}$ is the maximum width, and $\zeta$ (sampling rate, usually very small) is in the range $(0,1)$. Usually, we do not use a large $L_W$. Suppose $n=5000$, $L_W=4$, $\bar{d}=200$, and $\zeta=0.1$ (quite large), then $\frac{L_W^3 \xi_{\bar{d}}}{n}=0.256$, which is much less than 1. Indeed, it is possible that $\frac{L_W^3 \xi_{\bar{d}}}{n}$ is larger than 1 when $n$ is too small compared to the size of the neural network. But in practice, $n$ is usually very large.

---

### Meta-Review · Area_Chair_8KqR · 2023-12-06

**Metareview:**

This paper presents a neuron-enhanced auto-encoder matrix completion (AEMC-NE) method for collaborative filtering. The proposed method is well motivated, and the authors provide detailed theoretical justifications regarding the generalization ability. Reviewers raised some comments on technical details and experiments, which have been adequately addressed by the authors' responses. The authors are highly encouraged to incorporate the comments from reviewers into the final version.

**Justification For Why Not Higher Score:**

The justification of technical approaches should be further improved, as pointed out by reviewers.

**Justification For Why Not Lower Score:**

The paper presents a novel and effective method with theoretical justifications.

---

### Decision · Program_Chairs · 2024-01-16

Accept (poster)